# Deriving neural scaling laws from the statistics of natural language

**Francesco Cagnetta** [* 1]  **Allan Raventós** [* 2]  **Surya Ganguli** [† 2]  **Matthieu Wyart** [† 3 4]

## Abstract

Despite the fact that experimental neural scaling laws have substantially guided empirical progress in large-scale machine learning, no existing theory can quantitatively predict the exponents of these important laws for any modern LLM trained on any natural language dataset. We provide the first such theory in the case of data-limited scaling laws. We isolate two key statistical properties of language that *alone* can predict neural scaling exponents: (i) the decay of pairwise token correlations with time separation between token pairs, and (ii) the decay of the next-token conditional entropy with the length of the conditioning context. We further derive a simple formula in terms of these statistics that predicts data-limited neural scaling exponents from first principles *without any* free parameters or synthetic data models. Our theory exhibits a remarkable match with experimentally measured neural scaling laws obtained from training GPT-2 and LLaMA style models from scratch on two qualitatively different benchmarks, TinyStories and WikiText.

## 1. Introduction

How a language can be acquired from example sentences is a central question in linguistics and cognitive science. The *poverty of the stimulus* argument (Chomsky, 1980) challenged the very possibility of such learning, while distributional approaches have argued that regularities present in the input data may suffice (Ellis, 2002; Saffran et al., 1996; Saffran & Kirkham, 2018). The striking success of

[1]Theoretical and Scientific Data Science, International School for Advanced Studies (SISSA), Trieste, Italy [2]Department of Applied Physics, Stanford University, Stanford, California [3]Department of Physics and Astronomy, Johns Hopkins University, Baltimore, Maryland [4]Institute of Physics, École Polytechnique Fédérale de Lausanne (EPFL), Lausanne, Switzerland. Correspondence to: Allan Raventós <aravento@stanford.edu>, Francesco Cagnetta <fr.cagnetta@gmail.com>.

*Proceedings of the $43^{rd}$ International Conference on Machine Learning*, Seoul, South Korea. PMLR 306, 2026. Copyright 2026 by the author(s).

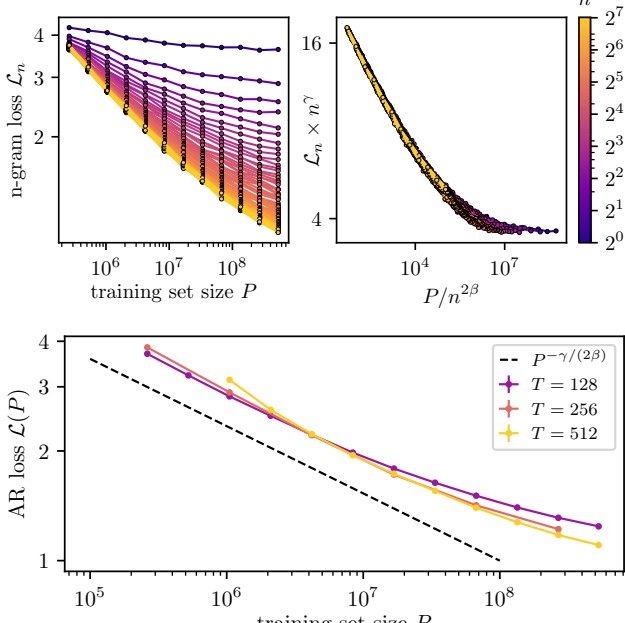

*Figure 1.* **Measurable language statistics predict the exponents of data-limited neural scaling laws in language models. Top:** The highly diverse $n$-gram losses $\mathcal{L}_n$ (Eq. 2) of a GPT-2–style transformer trained from scratch on $P$-tokens of the TinyStories dataset (left) collapse onto a *single* curve, when plotted in rescaled units (right). Here $\mathcal{L}_n$ is rescaled by $H_n \asymp n^{-\gamma}$, where $\gamma$ is the exponent of the power law temporal decay of $H_n$ with $n$, and $H_n$ is the next-token conditional entropy conditioned on the previous $n$ tokens. Also $P$ is rescaled by $n^{2\beta}$ where $\beta$ is the exponent governing the power-law decay of token-token correlations separated by temporal lag $n$. The entropy exponent $\gamma$ and the correlation exponent $\beta$ are strictly properties of the dataset, yet they completely control all neural $n$-gram learning curves $\mathcal{L}_n(P)$ through collapse. **Bottom:** We plot the autoregressive test loss $\mathcal{L}$ (which averages the $n$-gram losses over all $n$) as a function of $P$ for models trained with varying context size $T$. Our theory predicts that the exponent of the neural scaling law depends on language statistics alone via $\gamma$ and $\beta$ and is given by $\alpha_D = \gamma/(2\beta)$. Remarkably, our theoretical prediction (slope of the dashed black line) matches that of experimental neural scaling laws (colored lines), especially at larger context sizes $T$, as predicted by our theory.

Large Language Models (LLMs) (Radford et al., 2018; Devlin et al., 2019) has now decisively demonstrated that rich linguistic competence can emerge from exposure to data alone. Yet, despite this empirical success, the mechanisms and principles underlying such learning are not understood.

Beyond their qualitative abilities, LLMs exhibit quantitative regularities that are now central to both scientific inquiry and industrial practice. Most notably, their performance improves as a function of dataset size, model capacity, and compute budget, following approximate power laws known as *Neural Scaling Laws* (Hestness et al., 2017; Kaplan et al., 2020; Henighan et al., 2020; Hoffmann et al., 2022; Bahri et al., 2024). These scaling relations guide large-scale training decisions across the AI industry. Among these, the dependence of test loss on the amount of training data plays a particularly important role, as it directly governs the expected returns obtained from collecting additional data. Remarkably, what ultimately controls the exponent of the power law relating loss to data remains unknown.

Existing theoretical approaches to deriving loss learning curves have predominantly focused on kernel methods, where test loss can be predicted from the spectral properties of a *fixed* feature map or kernel. In this framework, learning-curve exponents are controlled by the alignment between the target function and the spectrum of the kernel (Caponnetto & De Vito, 2007; Spigler et al., 2020; Bordelon et al., 2020; Canatar et al., 2021; Loureiro et al., 2021; Bordelon et al., 2024). However, LLMs operate in a qualitatively different regime: during training, they learn to represent key features of languages, including syntactic (Peters et al., 2018; Tenney et al., 2019; Manning et al., 2020; Diego Simon et al., 2024) and semantic features (Engels et al., 2024; Gurnee et al., 2025; Acevedo et al., 2026). As a result, kernel-based theories with fixed features offer limited guidance for understanding the data-scaling behavior of modern feature-learning LLMs.

A different line of work has emphasized the latent hierarchical structure of data, studied via toy models of context-free grammars (Cagnetta et al., 2024; Cagnetta & Wyart, 2024; Allen-Zhu & Li, 2023; Garnier-Brun et al., 2025). A key emerging insight is that increasing the amount of data enables models to exploit progressively longer contexts, and that correlations between short substrings and future tokens can suffice to recover the latent hierarchical structure fully (Cagnetta & Wyart, 2024; Favero et al., 2025). Empirical evidence for this picture has been obtained in small-scale settings, at the character level and for very limited context lengths. Currently, this perspective faces two key limitations: first, it did not provide a direct prediction linking the statistics of *natural* language to the exponents of loss learning curves; second, it has not been tested at token scales relevant for modern LLMs.

In this work, we develop for the first time a general theoretical framework that *fully predicts the loss learning curve exponent from measurable statistical properties of natural language*. Our approach does not require any synthetic data model and has no free parameters that need to be fitted. Our

theory exposes two key statistical properties of language that *alone* determine loss learning curve exponents of LLMs: the decay of next token conditional entropy with context length, and the decay of token-token correlations with their temporal separation. We test our theory by carefully measuring these statistics in TinyStories and WikiText, and compare the resulting theoretical predictions for loss learning curve exponents to those of experimental learning curves obtained from GPT-2 and LLaMA-style transformers trained on these same datasets. Across architectures and datasets, we find a very good agreement, in the range of dataset sizes and context lengths probed, between the theoretically predicted and experimentally observed scaling behavior—both in terms of the power law exponent of the loss learning curve, and in terms of *scaling collapse* of families of loss curves with different fixed context lengths. An example of the striking match between our theory and experiment is illustrated in Fig.1. Overall, this work unravels, for the first time, a *direct* link between the shape of neural scaling laws and the statistical structure of language itself.

The code required for reproducing all experiments and figures is available at `https://github.com/fracagn etta/small-language-modelling`

## 1.1. Additional related works

**Empirical Scaling Laws.** The foundational work of (Kaplan et al., 2020) studies neural scaling laws in three regimes: model-size-limited, data-limited, and compute-limited, and empirically observes power-law decays in each. (Hoffmann et al., 2022) revised the compute-optimal training prescription of (Kaplan et al., 2020), with several subsequent works studying the origins and robustness of Chinchilla scaling (e.g., (Porian et al., 2025; Schaeffer et al., 2025)). In this work, we focus on the data-limited regime. Thus, our empirics operate in a setting in which model size and compute do not constrain performance but data amount does, similar to (Kim et al., 2025), which requires extensive experiments.

**Solvable models.** Several works consider synthetic data distributions in which scaling exponents can be obtained analytically. Many focus on linear models; for example, (Spigler et al., 2020; Sharma & Kaplan, 2020) relate test error in regression to a dataset's intrinsic dimension; (Maloney et al., 2022) solve a joint generative and random feature model; (Lin et al., 2024) analyze linear regression in the asymptotic joint model and dataset size limit; and (Paquette et al., 2024) derive phase structure and compute-optimal tradeoffs. Other works consider discrete data distributions; (Hutter, 2021) derives scaling exponents for tabular learning under various feature distributions. (Kunstner & Bach, 2025) obtains time-dependent scaling exponents for linear bigram models trained on Zipf-distributed tokens, while (Yüksel et al., 2025) shows that high-order Markov chains where

the dependence on the past decreases with the time lag are learned sequentially. (Sorscher et al., 2022) show how to achieve better than power-law scaling in both theory and practice through data curation.

**Power laws in data distributions.** (Michaud et al., 2023) posit that power-law scaling in test error arises from Zipf-distributed quanta in data. This view was proved for associative memories (Cabannes et al., 2024), for the multitask sparse parity problem (Nam et al., 2024), and regression with labels generated by a two-layer network, whose neurons act as quanta (Ren et al., 2025). (Debowski, 2025) proposes a toy model of sequences where Zipf-distributed token frequencies can induce a power-law decay of the next-token cross-entropy. However, the construction is highly stylized, and the extent to which it captures long-range dependencies in language remains unclear. In contrast, (Barkeshli et al., 2026) observe power-law scaling even when the underlying data do not exhibit explicit power-law structure, considering synthetic settings such as random walks on graphs. Similarly, Liu et al. (2025) show that representation superposition can induce power-law scaling behavior even when no such structure is present in the input features. In contrast, our work directly relates power-law scaling in test error to two intrinsic properties of *natural* language.

## 2. Notation and setup

We consider the auto-regressive training of language models on a text corpus. Formally, a corpus defines a distribution over sequences of tokens, $(x_1, x_2, \ldots)$, where each $x_i$ belongs to a finite vocabulary $\mathcal{V}$. We denote the unknown underlying distribution over sequences by $\mathbb{P}$ and write $p_n(x_{1:n}) \equiv \mathbb{P}(X_1 = x_1, \ldots, X_n = x_n)$ for its marginal over length-$n$ sequences. In practice, the corpus is a concatenation of documents, from which contiguous subsequences of length $T + 1$ are sampled to form a dataset $\mathcal{D}$. We will denote by $P$ the total number of *tokens* in $\mathcal{D}$.

A language model $\hat{p}_\theta$ with parameters $\theta$ defines, for $n$ ranging from 1 to the maximum context size $T$, a conditional distribution, $\hat{p}_\theta(x_{n+1} \mid x_{1:n})$, of the next token, $x_{n+1}$, conditioned on all previous tokens in the context $x_{1:n} \equiv (x_1, \ldots, x_n)$. We refer to $n$ as the *time horizon*. Given a dataset $\mathcal{D}$, training proceeds by gradient descent on the negative log-likelihood,

$$\mathcal{L}(\theta) = -\mathbb{E}_{x_{1:T+1} \sim \mathcal{D}} \left[ \frac{1}{T} \sum_{n=1}^{T} \log \hat{p}_\theta(x_{n+1} \mid x_{1:n}) \right] \quad (1)$$

We denote by $\mathcal{L}_n$ the $n$-gram loss,

$$\mathcal{L}_n = -\mathbb{E} \left[ \log \hat{p}_\theta(x_{n+1} \mid x_{1:n}) \right] \quad (2)$$

such that $\mathcal{L}(\theta) = \frac{1}{T} \sum_{n=1}^{T} \mathcal{L}_n$. $\mathcal{L}_n$ achieves its minimum at the *conditional entropy* $H_n = \mathbb{E} \left[ -\log \mathbb{P}(x_{n+1} \mid x_{1:n}) \right]$,

which only depends on the dataset $\mathcal{D}$ and is a decreasing function of the time horizon $n$.

## 3. Theory: data-limited scaling exponents from natural language statistics

A detailed derivation of our theory is provided in App. A. Here we just sketch the salient aspects. Our theory is focused on the data-limited scaling exponent, describing the power law reduction of the autoregressive loss in Eq. 1 with an increasing amount of training data. The theory is based on a simple decomposition of the loss into two sources of error. The first is due to an effective *prediction time horizon*, i.e. how many previous tokens can the language model actually take into account when predicting the next token. The second is due to how suboptimally the language model uses information *within* the prediction time horizon to predict the next token. Consequently, there are two learning mechanisms: increasing the prediction time horizon, and improving the prediction within the horizon. As we show in the following (details in App. A), when the first mechanism dominates, we can predict the data-limited scaling exponent from statistical properties of the dataset alone.

**Data-dependent prediction time horizon.** Given the number of training tokens $P$, the data-dependent prediction time horizon $n^*(P)$ can be defined as the maximal context window size that the model can beneficially leverage for next token prediction (Cagnetta & Wyart, 2024). In essence, the model can only use tokens up to $n^*(P)$ timesteps in the past. The minimal requirement for beneficial use would be that the amount of data $P$ is large enough that one can detect the strongest token-token correlations at temporal separation $n$. We measure this dependency via the two-point token-token covariance matrix over the whole dataset,

$$C_{\mu,\nu}(n) = \mathbb{P}\{X_i = \mu, X_{i+n} = \nu\} - \mathbb{P}\{X_i = \mu\} \mathbb{P}\{X_{i+n} = \nu\}, \quad (3)$$

and take the top singular value $\|C(n)\|_{\mathrm{op}}$ as a measure of the strongest signal in the matrix. Note that, since we consider a setup where $P$ increases while the vocabulary size stays fixed, taking the first few singular values, or measuring the signal via the Frobenius norm, yields the same scaling. Limiting the measure of covariance to the $P$ training tokens induces an *additive* sampling noise $\widehat{C}_P(n) - C(n)$ on each matrix element. The noise is $O(P^{-1/2})$, as expected from a central-limit-theorem scaling—see App. B for details. Comparison with the signal $\|C(n)\|_{\mathrm{op}}$ gives the threshold,

$$\|C(n)\|_{\mathrm{op}} = O\left(\frac{1}{\sqrt{P}}\right), \quad (4)$$

which yields $n^*(P)$ after solving for $n$. We expect $n^*(P)$ to grow with $P$ (e.g. more training data allows the model to beneficially use tokens further back in the past to predict).

**Loss decomposition.** Given $n^*(P)$, the next-token conditional entropy at the prediction time horizon $H_{n^*(P)}$ bounds from below all the $n$-gram losses contributing to the autoregressive loss. Any loss above this lower bound can be attributed to the *suboptimal* use of tokens within the prediction time horizon, with $n < n^*(P)$. When the maximal context length satisfies $T \gg n^*(P)$ (otherwise the loss is limited by the maximal context), we decompose the loss as

$$\mathcal{L}_{\mathrm{AR}}(P) \asymp H_{n^*(P)} + \sum_{n=1}^{n^*(P)} \mathcal{E}_n(P), \qquad (5)$$

where $\mathcal{E}_n(P)$ measures the suboptimal use of the $n$-th past token (see App. A for a derivation). The terms in the decomposition represent the two aforementioned learning mechanisms: increasing the prediction time horizon $n^*(P)$, which reduces $H_{n^*(P)}$, or improving the prediction within the horizon, which reduces the $\mathcal{E}_n(P)$'s.

**Language statistics.** To turn Eq. 5 into a prediction of the data-limited scaling law, we resort to two empirically-motivated hypotheses about the decay of the next token conditional entropy $H_n$ with the length $n$ of the conditioning context, and the decay of token-token correlation strength $\|C(n)\|_{\mathrm{op}}$ with the time separation $n$. Namely,

$$H_n - H_\infty \asymp n^{-\gamma}, \qquad (6)$$

$$\|C(n)\|_{\mathrm{op}} \asymp n^{-\beta}. \qquad (7)$$

Eq. 6 is a differential form of Hilberg's hypothesis (Hilberg, 1990; Crutchfield & Feldman, 2003; Takahira et al., 2016) for how the scaled total entropy, $T^{-1}\sum_{n=1}^{T} H_n$, approaches the *entropy rate* $H_\infty$. Synthetic hierarchical models of language data (Cagnetta & Wyart, 2024) satisfy this hypothesis. The power-law decay of correlations in Eq. 7 can also be attributed to the hidden hierarchical structure of language (Lin & Tegmark, 2017; Cagnetta & Wyart, 2024).

**Data-limited scaling exponent.** Now Eq. 4 and Eq. 7 yield $n^*(P) \asymp P^{1/(2\beta)}$ for the data-dependent prediction time horizon, and reveals that it grows as a power law with the amount of data $P$. Then, as detailed in App. A, plugging these hypotheses into Eq. 5 and assuming that the $\mathcal{E}_n(P)$'s decay with $P$ faster than $H_{n^*(P)}$, yields

$$\mathcal{L}_{\mathrm{AR}}(P) - H_\infty \asymp P^{-\frac{\gamma}{2\beta}}. \qquad (8)$$

Eq. 8 captures a *horizon-limited* regime in which the dominant contribution to the excess test loss comes from the fact that, at dataset size $P$, only statistical token dependencies up to a maximal time separation $n^*(P)$ are available to aid prediction. The assumption that the $\mathcal{E}_n(P)$ are not dominant implies that most of the language structure on context scales $n \ll n^*(P)$ is already acquired by a model trained with $P$ data. Whether this assumption of fast learning within $P$ data. Whether this assumption of fast learning within

the time horizon holds is clearly architecture-dependent, as demonstrated in controlled synthetic datasets where $\gamma$ and $\beta$ can be manipulated independently (Cagnetta et al., 2024; Cagnetta & Wyart, 2024). In particular, we expect this assumption to fail in shallow networks, kernel methods or $n$-gram models: as the context dimension increases with $n$, these methods would suffer from the curse of dimensionality, preventing fast learning within the context. Our assumption may instead apply to a class of deep networks that includes the LLMs considered in this paper, as our experiments below directly indicate.

**Collapse of the $n$-gram losses.** A more general implication of our framework is obtained by considering the *individual* $n$-gram learning curves $\mathcal{L}_n(P)$. When plotted against the dataset size $P$, these curves need not coincide, since each horizon $n$ becomes usable only after a time-dependent data threshold $P_n^*$—the minimal amount of data required for the model to reliably leverage tokens at separation $n$. This suggests reparameterizing each curve by the rescaled data amount $\bar{P}_n \equiv P/P_n^*$. Moreover, $\mathcal{L}_n(P)$ has a natural vertical scale given by its asymptote $H_n = \lim_{P\to\infty} \mathcal{L}_n(P)$, motivating the normalized loss $\ell_n(\bar{P}_n) \equiv \mathcal{L}_n(P)/H_n$ evaluated at $\bar{P}_n$. Under the scaling ansätze in Eq. 6 and Eq. 7, this yields the scaling form (derived in App. A)

$$\mathcal{L}_n(P) \equiv n^{-\gamma} \ell\left(P/n^{2\beta}\right). \qquad (9)$$

Therefore, once the two exponents $\gamma$ and $\beta$ are measured, the different $n$-gram learning curves, plotted in the rescaled variables, should approximately collapse onto a single master curve $\ell$. This collapse is indeed observed in our experiments, providing a stringent validation of our scaling theory.

## 4. Empirical verification on text corpora

### 4.1. Estimating language statistics from text

The first step in the validation of our theory is the measurement of the two language exponents that conspire to determine the data-limited learning curve scaling exponent: $\gamma$, governing the decay of the next-token conditional entropy with conditioning time horizon in Eq. 6; and $\beta$, governing the temporal decay of token-token correlations in Eq. 29. We consider two qualitatively different datasets: TinyStories (a collection of short stories generated by GPT-3.5 and -4 (Ronen & Yuanzhi, 2023)) and WikiText-103 (a collection of verified articles from Wikipedia (Merity et al., 2017)).

#### 4.1.1. CONDITIONAL ENTROPIES AND EXPONENT $\gamma$

Direct estimation of the conditional entropies $H_n$ from raw counts is computationally infeasible at the vocabulary sizes and horizons of interest. Indeed, since the number of distinct contexts of length $n$ grows exponentially with $n$, reliable frequency estimates would require prohibitive sample sizes. To

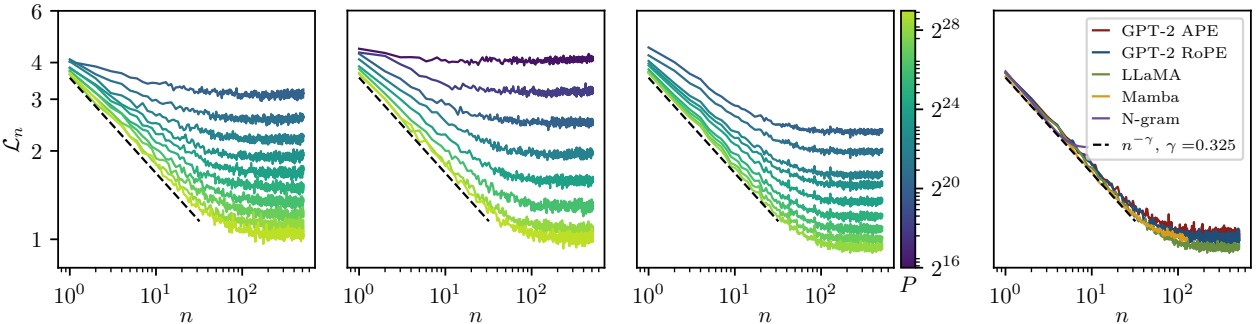

*Figure 2.* **Conditional entropy decay with time horizon defines a characteristic exponent $\gamma$ that is architecture-independent.** We train three classes of models from scratch on $P$-token slices of the TinyStories dataset: GPT-2–style transformers with absolute positional embeddings (**Left**), GPT-2–style transformers with rotary positional embeddings (RoPE, **Center Left**), and LLaMA-style transformers (**Center Right**), training a separate model for each $P$. For each model class, we measure the $n$-gram loss $\mathcal{L}_n$ as a function of $n$, with curves colored by $P$. We define $\gamma$ by fitting a power law to the initial decay of $\mathcal{L}_n$ for the model trained on the largest $P$. As $P$ increases, the small-$n$ region of $\mathcal{L}_n$ starts to converge, and the fitted exponent stabilizes. Crucially, the resulting values of $\gamma$ are consistent across architectures (**Right**). Thus, for a given dataset, $\gamma$ is a property of the data distribution and can be estimated from a single sufficiently large, well-trained model.

circumvent this issue, we use trained autoregressive models.

Concretely, given a number of training tokens $P$ and a model family $\mathcal{M}$, training yields the language model (parameter subscript $\theta$ omitted for clarity) $\hat{p}_{P,\mathcal{M}}(x_{n+1} \mid x_{1:n})$ for all $n \leq T$. The corresponding $n$-gram loss $\mathcal{L}_n(P, \mathcal{M})$, from Eq. 2, bounds the conditional entropy from above as

$$\mathcal{L}_n(P, \mathcal{M}) \geq H_n, \quad (10)$$

with equality achieved in the infinite data limit, given sufficient capacity of the model class. Our strategy is to treat, for a fixed expressive model class $\mathcal{M}$ and increasing $P$, $\mathcal{L}_n(P, \mathcal{M})$ as a sequence of increasingly accurate upper bounds on $H_n$. Empirically, we observe that the $n$-gram losses indeed converge with increasing $P$ towards a limiting curve, especially for small time horizons $n$. [1] Results are shown in Fig. 2 for TinyStories and in Fig. 5 for WikiText. This observation motivates estimating $\gamma$ from a power-law fit of the small-$n$ portion of the $\mathcal{L}_n(P, \mathcal{M})$-v-$n$ curve for the largest $P$ available, yielding

$$\text{TinyStories:} \quad \gamma = 0.325 \pm 0.003, \quad (11)$$
$$\text{WikiText:} \quad \gamma = 0.265 \pm 0.016. \quad (12)$$

The errors are obtained via a bootstrapping procedure detailed in App. C. Note that our exponents are slightly larger than the 0.23 found by (Takahira et al., 2016), but they used different data (mostly news articles), character-level tokens, and estimated entropies via compression.

To verify that the limiting behavior is not an artefact of a particular architecture, we repeat the procedure with

both additional transformer-based (GPT-2 with rotational positional encoding and LLaMA) and non-transformer (infini-gram (Liu et al., 2024) and Mamba (Gu & Dao, 2024), details in App. E) language models. The agreement of the limiting $\mathcal{L}_n$-v-$n$ curves across all classes, together with the convergence for increasing number of tokens, confirms that the decay of the conditional entropy is a property of the dataset—namely $H_n$—rather than of the specific model. Computing these exponents is one of our central results, revealing a new quantitative statistical property of language.

### 4.1.2. TOKEN-TOKEN CORRELATIONS AND EXPONENT $\beta$

The statistical dependence between tokens with a time lag $n$ is directly quantified by the $V \times V$ covariance matrix from Eq. 3. We extract a scalar summary of the correlation strength by taking the operator norm $\|C(n)\|_{\text{op}}$, that is, the largest singular value of the covariance matrix $C(n)$. Then, following Eq. 7, we extract $\beta$ from a power-law fit of the $\|C(n)\|_{\text{op}}$-v-$n$ curve. The results, as reported in Fig. 3, are

$$\text{TinyStories:} \quad \beta = 0.88 \pm 0.06, \quad (13)$$
$$\text{WikiText:} \quad \beta = 0.94 \pm 0.16, \quad (14)$$

where, as for entropies, the errors are obtained via bootstrapping. Note that, empirically, the spectra of the covariance matrices are broad, i.e., a small number of singular directions capture a large fraction of the matrix. Consequently, the decay of the correlation strength with $n$ is not tied to a specific choice of norm, and using the Frobenius norm $\|C\|_{\text{F}} = (\sum_{i,j} C_{i,j}^2)^{1/2}$ yields the same decay.

In addition, the decay of correlations in WikiText (right panel of Fig. 3) is better described as a broken power law with two stages. We use the exponent of the short-lag stage $n \lesssim 32$, which is the range where $n^*(P)$ falls, given the

---

[1] (Scheibner et al., 2025) also uses LLMs for estimating entropies, but on corpora different from the training dataset. One cannot expect convergence to the entropy of the training distribution in this case.

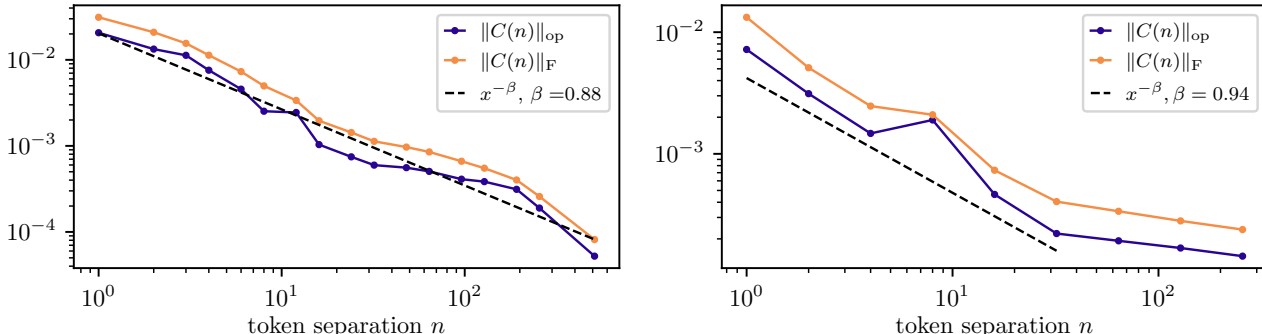

*Figure 3.* **Decay of two-point correlations as a function of temporal separation defines a characteristic exponent $\beta$.** The second dataset-level statistic we consider is the decay of the two-point correlation function, defined as the norm of the token-token co-occurrence matrix $C(n)_{\mu\nu} = \mathbb{P}(X_i = \mu, X_{i+n} = \nu) - \mathbb{P}(X_i = \mu)\mathbb{P}(X_{i+n} = \nu)$, with the time separation $n$. We define $\beta$ as the exponent of a power-law fit to this decay. $C(n)$ is estimated from empirical token co-occurrence counts on the full training set for TinyStories (**Left**) and WikiText (**Right**). We plot both the Frobenius and operator norms, which closely track each other, and use the operator norm to characterize the decay. For TinyStories, the power law holds over a broad range of time separations, while for WikiText we fit $\beta$ using the initial decay regime.

considered range of $P$. This is visible in Fig. 4, top, where the $n$-gram learning curves with $n > 32$ do not bend significantly. We also observed a localized peak, around $n \approx 10$, which also appears for TinyStories, although not as prominently. As our goal is a coarse-grained scaling description that is intentionally insensitive to such details, we do not take this outlier peak into account.

### 4.2. Empirical scaling in trained models

Having measured the two dataset-level exponents $\gamma$ (conditional entropy decay) and $\beta$ (correlation decay) in subsection 4.1, we now test our central prediction: once the within-horizon excess losses decay sufficiently rapidly, the data-limited learning-curve exponent $\alpha_D$ should be set purely by language statistics, $\alpha_D = \gamma/(2\beta)$, with no additional fitting parameters. We validate this prediction both by testing the collapse of the individual $n$-gram learning curves under the rescaling of Eq. 9, and by verifying that the full autoregressive learning curve exhibits the predicted power-law decay across varying maximal context lengths $T$. We discuss error estimates and confidence intervals in subsubsection 4.2.3. Implementation details are provided in App. E.

#### 4.2.1. TINYSTORIES

Fig. 1 (top left) shows the family of $n$-gram losses $\mathcal{L}_n(P)$ obtained by training a fixed GPT-2-style APE transformer from scratch on $P$ tokens of TinyStories, for a range of time horizons $n$. As expected, these curves do *not* lie on top of each other in the raw $(P, \mathcal{L})$ plane: larger $n$ corresponds to a harder conditional prediction problem, hence to a higher data requirement for improvements to become visible. The theory asserts that both effects have natural $n$-dependent scales: vertically, the relevant loss scale is the

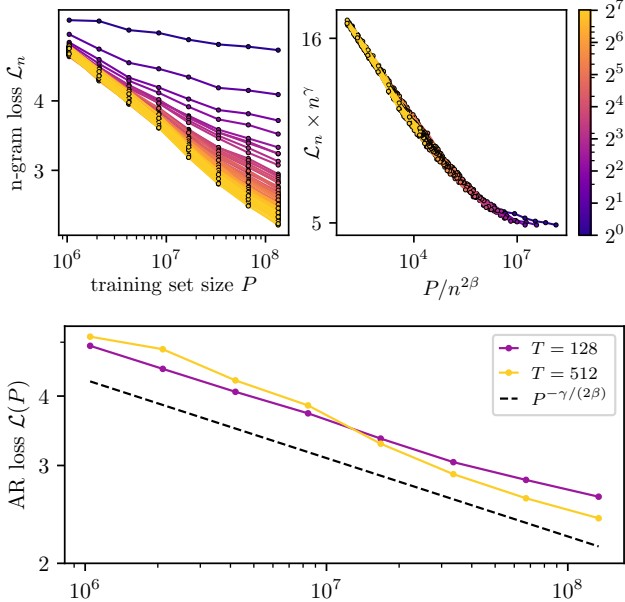

*Figure 4.* **Same as Fig. 1, but for GPT-2–style transformers trained on WikiText.** As in Fig. 1, the top row is for $T = 128$; see Fig. 13 for $T = 512$. Corresponding figures for GPT-2–style transformers with RoPE on WikiText are in Fig. 14.

dataset's conditional entropy $H_n$ and its power-law decay $H_n - H_\infty \asymp n^{-\gamma}$; horizontally, the relevant data scale is the time-dependent data threshold $P_n^*$ at which token-token correlations at lag $n$ become resolvable, predicted to scale as $P_n^* \asymp n^{2\beta}$. Fig. 1 (top right) shows the same curves in the rescaled variables

$$P \mapsto P/n^{2\beta}, \qquad \mathcal{L}_n \mapsto n^\gamma \mathcal{L}_n,$$

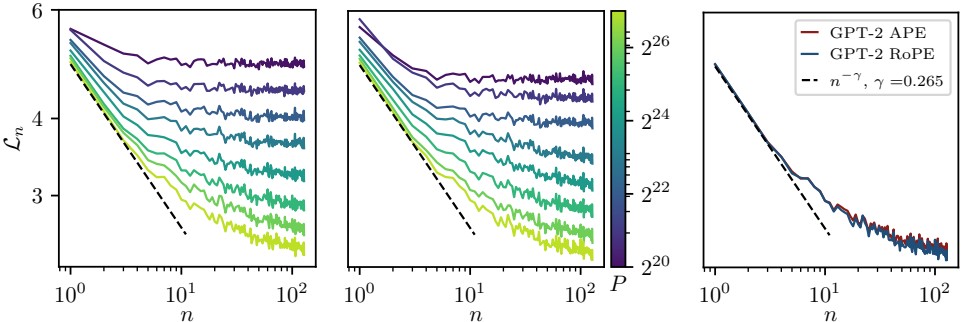

*Figure 5.* **Fitting $\gamma$ for the WikiText dataset.** (Similar to Fig. 2, but for WikiText.) We train two classes of models from scratch on $P$-token slices of the WikiText dataset: GPT-2–style transformers with APE (**Left**) and GPT-2–style transformers with RoPE (**Center**). For each model class, $\gamma$ is obtained by fitting a power law to the initial decay of $\mathcal{L}_n$ for the model trained with the largest $P$. As in Fig. 2, the resulting estimates of $\gamma$ are consistent across architectures (**Right**).

and reveals a striking collapse of *all* curves onto a *single curve* across the whole range of $n$ and $P$. This collapse is a direct signature of our proposed learning mechanism: increasing $P$ primarily extends the prediction time horizon $n^*(P)$ rather than slowly refining performance at fixed $n$.

The same mechanism yields a quantitative prediction for the full autoregressive loss $\mathcal{L}(P)$. Fig. 1 (bottom) shows $\mathcal{L}(P)$ for models trained with varying maximal context length $T$. All the curves exhibit a common power-law decay over a broad range of $P$ such that the finite context length does not truncate the data-dependent horizon (i.e. $n^*(P) \ll T$). The observed decay is consistent with the predicted exponent $\alpha_D = \gamma/(2\beta)$. Using the measured TinyStories values $\gamma = 0.325$ and $\beta = 0.88$ gives the prediction

$$\alpha_D \approx \frac{0.325}{2 \times 0.88} \approx 0.185,$$

As indicated by the dashed black line in Fig. 1 (bottom), our prediction matches empirical learning curves. Importantly, the consistency across several $T$ values supports the fast-learning assumption, i.e. that the dominant benefit of additional data $P$ is to unlock progressively longer useful context (increasing $n^*(P)$), rather than to substantially reduce the within-horizon excess losses at fixed $n < n^*(P)$.

### 4.2.2. WIKITEXT

Fig. 4 repeats the same analysis on WikiText. Despite the difference in datasets (including the more complex structure of correlations discussed in subsection 4.1), the two empirical signatures persist. First, $n$-gram learning curves collapse again under the same rescaling (top panels). Second, the autoregressive loss $\mathcal{L}(P)$ (bottom) follows a power law compatible with the predicted exponent $\alpha_D = \gamma/(2\beta)$ across different context lengths $T$. Plugging in the measured WikiText exponents $\gamma = 0.27$ and $\beta = 0.94$ yields

$$\alpha_D \approx \frac{0.265}{2 \times 0.94} \approx 0.141,$$

consistent with the empirical scaling shown in Fig. 4.

Overall, Figs. 1 and 4 provide convergent evidence for the theory: (i) the rescaled collapse supports the existence of an $n$-dependent data threshold and entropy normalization governing $\mathcal{L}_n(P)$, and (ii) the resulting exponent prediction $\alpha_D = \gamma/(2\beta)$ quantitatively matches the observed autoregressive scaling across two disparate corpora, with no fitted parameters beyond independently measured language statistics.

### 4.2.3. ERROR ESTIMATES

The error on our predicted $\alpha_D$ can be estimated via standard propagation of uncertainty (see App. C),

$$\text{TinyStories: } \alpha_D = 0.185 \pm 0.013, \tag{15}$$
$$\text{WikiText: } \alpha_D = 0.141 \pm 0.025. \tag{16}$$

We can compare these intervals with empirical exponents $\widehat{\alpha}_D$ extracted directly from empirical learning curves. There are two practical limitations in extracting such exponents. First, the maximal context window $T$ imposes a lower bound on the number of tokens in a batch, making the small-$P$ region of curves with large $T$ harder to optimize. Second, for sufficiently large $P$ at fixed $T$, our prediction is expected to break down as $n^*(P)$ approaches $T$. We therefore compute the lower envelope of learning curves across different values of $T$, and estimate the exponent by fitting the first $m$ points of this envelope. Using bootstrap resampling to obtain 95% confidence intervals, we find overlap with the predicted range for up to 10 of the first 12 points on TinyStories, and for all 8 points on WikiText. See App. C for details.

We also examine how the uncertainty in $\beta$ and $\gamma$ affects the scaling collapse of the $n$-gram learning curves $\mathcal{L}_n(P)$. As shown in App. C, for TinyStories the collapse visibly deteriorates when $\beta$ is taken outside its standard-error interval Eq. 13, and when $\gamma$ is taken outside the range $[0.31, 0.34]$.

The latter is wider than the standard-error interval reported in Eq. 11, but contains the fitted exponents obtained when varying the model class and the fit range. An exception is the Llama model trained on TinyStories, for which the collapse appears visually sharper for $\beta \in [0.6, 0.7]$, below the standard-error interval. Notably, however, the corresponding scaling law remains compatible with the predicted exponent $\alpha_D$, so this discrepancy does not affect our main conclusions and we leave its interpretation for future work. For WikiText, we find that the collapse deteriorates qualitatively when $\beta$ is taken outside its standard error interval Eq. 14, and when $\gamma$ is taken outside the range $[0.23, 0.30]$.

## 5. Improvement within the prediction time horizon

In this section, we present further empirical evidence supporting the horizon-limited regime, where the decay of the excess losses $\mathcal{E}_n(P)$ is faster than $P^{-\gamma/(2\beta)}$.

While the $\mathcal{E}_n(P)$ are not easily accessed, we can test the assumption directly on the $n$-gram learning curves $\mathcal{L}_n(P)$. These curves indeed decrease via the two same mechanisms affecting the autoregressive loss: increase of the prediction time horizon and/or better use of information within the horizon—see Eq. 43 for a derivation. The effect of increasing the prediction time horizon saturates as soon as $n^*(P)$ becomes comparable with $n$, which we can identify by tracking the distance between the top singular values of the empirical token-token covariance at time lag $n$ estimated with $P$ tokens, and those of the true (full dataset) covariance matrices. For small-to-intermediate $n$, saturation happens at relatively small $P$. The remaining large $P$ section of the $n$-gram learning curve $\mathcal{L}_n(P)$ is entirely controlled by the suboptimal use of tokens within the time horizon $n < n^*(P)$, hence we can use it to extract an empirical estimate of the exponent $\delta_n$ controlling the asymptotic decay of $\mathcal{E}_n(P)$ via Eq. 26.

In practice, we select the $n$-gram losses with $n \leq 12$ of a language model with fixed maximal context window $T$ with varying $P$. For each loss, we take the points with $P$ at least $10\times$ larger than $P^*(n)$, and estimate $H_n$ as follows: for each value $H$ on a grid with step $10^{-2}$, fit a linear regression model to $\log(\mathcal{L}_n - H)$-v-$\log n$, then select the $H$ maximising the coefficient of determination of the fit. The maximising fit also yields $\delta_n$ as the slope. The results are presented in Fig. 6 for GPT-2-style transformer with $T = 128$ trained on Tinystories. The figure (in particular, the right panel, which collects the exponents of all the $n$'s considered) shows that the large-$P$ portion of the $n$-gram learning curves decays faster than the autoregressive loss scaling prediction $P^{-\gamma/(2\beta)}$.

This indicates further evidence for the fast learning horizon

limited regime: as data amount $P$ increases, the model can detect token-token correlations over an increasing maximal prediction time horizon of $n^*(P) \asymp P^{1/(2\beta)}$, and it very quickly learns to use all tokens *within* this time horizon to predict well. Thus the maximal prediction time-horizon and next-token conditional entropy at that conditioning time alone control the loss as a function of $P$, yielding Eq. 8.

## 6. Conclusions

Despite the considerable practical impact of scaling laws, a quantitative theoretical understanding of their underlying mechanisms has remained elusive. Here, we propose a theory for the data-limited learning curve exponents of LLMs trained on natural language, elucidating how the autoregressive loss depends jointly on context length and dataset size. Remarkably, all parameters entering our theory can be inferred directly from empirical language statistics, enabling stringent tests of its predictions.

In physics, phase transitions are characterized by universality classes: large sets of systems that share the same critical exponents. This notion is also central in the present context: which model architectures and training dynamics give rise to the scaling exponent and behavior we propose? The inability of kernel methods or shallow networks to learn even simple toy models of context-free grammars (Cagnetta et al., 2024) suggests that they lie outside this universality class, and exhibit much worse exponents than those reported here. An interesting possibility to explore is that a broad class of deep architectures — potentially including state-space models and even CNNs — do fall into the universality class discovered in this paper, which corresponds to a horizon-limited learning mechanism in which the loss is dominated by the finiteness of data-dependent prediction time horizon, rather than by the suboptimality of predictions *within* this time horizon. If so, the broad class of architectures falling into this universality class would share the same (data-set dependent) power law exponents for their learning curves, across all datasets, while differing perhaps substantially in their non-universal prefactors.

Lastly, our work seems to indicate a fundamental limit to the performance that LLMs can achieve that is solely determined by the statistical structure of the dataset. For example, at first sight it would seem unlikely that the structure of language can be learned in a limited data (small $P$) and large horizon/context scale (large $n$) regime where even pairwise correlations do not yet emerge from noise. However, this guess is not true in all generality: in simple toy models of hierarchical data, although the performance of transformers trained on predicting the last token of a sequence agrees with the present framework, CNNs can actually perform better thanks to translational equivariance (Cagnetta et al., 2025). Although this observation may not hold for natural

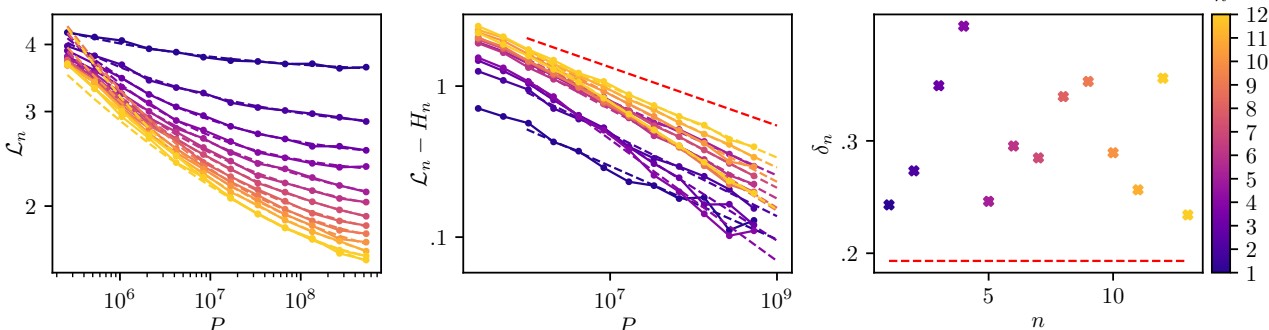

*Figure 6.* **$n$-gram losses decay to their asymptote faster than $P^{-\gamma/(2\beta)}$. Left:** We plot the $n$-gram losses of our GPT-2–style transformer trained on TinyStories for low $n$ ($n \leq 12$). To isolate the decay due to the suboptimal use of the time horizon, we fit the large-$P$ portion of these curves to the decay-to-asymptote form $A \times P^{-\delta_n} + H_n$. We perform a grid search over asymptotes with a step of $10^{-2}$, then subtract the asymptote and fit a linear regression model to the shifted data on the logarithmic scale. We then select $H_n$ as the value maximizing the coefficient of determination $R^2$ of the fit. **Middle:** The power-law decay of $\mathcal{L}_n - H_n$ with $P$ is compared with $P^{-\gamma/2\beta}$, shown as a red dashed curve. $\mathcal{L}_n - H_n$ decays faster for all $n$'s. **Right:** Scatter plot of the exponents $\delta_n$ for $n \leq 12$, clearly showing that $\delta_n > \gamma/(2\beta)$ (red dashed).

languages trained on next-token prediction, it suggests the possibility of an as-yet undiscovered universality class of architectures and learning algorithms that could exhibit superior data-limited learning curves with larger exponents than those theoretically derived and empirically measured here.

## Limitations

A key practical subtlety is that our tests necessarily probe a *finite-range* scaling regime set by both the largest dataset size $P_{\max}$ and the maximal context length $T$. At fixed $P_{\max}$, the largest effective horizon $n^*(P_{\max})$ that can be unlocked is limited, so that many of the horizons $n$ that we probe are not yet in a saturation/bending regime. In our experiments, $n^*(P_{\max})$ corresponds to a few tens of tokens—see, for instance, the saturation of $\mathcal{L}_n$ in Fig. 2 (right). Therefore, our theory is tested at the level of a few sentences, which already encompasses rich linguistic structure, including syntax. Concerning $T$, the most pressing question is whether the "horizon-limited" abstraction is plausible at current state-of-the-art scales, with dataset sizes well over the trillion scale and context window sizes $\gtrsim 10^5$. In general, it would be interesting to determine whether our conclusions extend to the larger maximal context lengths and dataset sizes accessible in industrial settings. We hope that the present study, obtained at an academic scale, will motivate such investigations.

## Acknowledgements

The work of FC was supported by the European Union's Horizon Europe program under the Marie Skłodowska-Curie grant agreement No. 101154584. SG thanks a Schmidt Sciences Polymath Award for support. This work was supported by the Simons Foundation through the Simons Collaboration on the Physics of Learning and Neural Computation (Award ID: SFI-MPS-POL00012574-05), PIs Ganguli and Wyart.

## Impact Statement

This paper presents work whose goal is to advance the field of Machine Learning. There are many potential societal consequences of our work, none which we feel must be specifically highlighted here.

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

## A. Theory of asymptotic scaling of the autoregressive loss

We consider an autoregressive model that, given a sequence of $n$ input tokens $(x_1, \ldots, x_n) \equiv (x_{1:n})$, approximates the true conditional probability of $X_{n+1}$ given $(X_1 = x_1, \ldots, X_n = x_n)$ for all $n = 1, \ldots, T$,

$$p_n(x|x_{1:n}) := \mathbb{P}\{X_{n+1} = x | X_1 = x_1, \ldots, X_n = x_n\}, \tag{17}$$

where $T$ is the maximal possible context length allowed by the model. We denote the model's approximation of these true conditionals $p_n$ by the approximate conditionals $\widehat{p}_n$. The auto-regressive loss $\mathcal{L}_{\text{AR}}$ is the mean of the cross-entropies between the model prediction $\widehat{p}_n$ and the data distribution $p_n$ over all past times $n$ within the maximal context window $T$:

$$\mathcal{L}_{\text{AR}} = \frac{1}{T} \sum_{n=1}^{T} \mathbb{E}_{(X_{1:n+1}) \sim p} \left[ -\log \widehat{p}_i \left( X_{n+1} \Big| X_{1:n} \right) \right] \equiv \frac{1}{T} \sum_{n=1}^{T} \mathcal{L}_n. \tag{18}$$

By replacing, for all $n$, $\widehat{p}_n$ with $\widehat{p}_n/p_n \times p_n$,

$$\mathcal{L}_{\text{AR}} = \frac{1}{T} \sum_{n=1}^{T} \Big( H\left(X_{n+1}|X_{n:i}\right) + \mathbb{E}_{x_{1:n}} \left[ D_{KL}\left(p_i(X_{n+1}|x_{1:n})||\widehat{p}_i(X_{n+1}|x_{1:n})\right) \right] \Big). \tag{19}$$

Here $D_{KL}(\cdot|\cdot) \geq 0$ is the KL divergence, which is nonnegative and $0$ if and only if its two arguments are the same distribution. Also $H\left(X_{n+1}|X_{1:n}\right)$ denotes the $n$-gram conditional entropy, or the conditional entropy of the next token given the past $n$ past tokens. This conditional entropy is strictly a property of the true data distribution, not the model, and is

$$H_n := H\left(X_{n+1}|X_{1:n}\right) = \mathbb{E}_{x_{1:n}} \left[ -\sum_x p_n(x|x_{1:n}) \log p_n(x|x_{1:n}) \right]. \tag{20}$$

If training progresses successfully, then the model approximations $\widehat{p}_n$ converge to the true distributions $p_n$, so the KL divergence term goes to $0$. Thus the $n$-gram losses $L_n$ converge to the corresponding conditional entropies $H_n$ from above, and the entire autoregressive loss converges to the entropy rate of the data $\frac{1}{T} H(X_{1:T+1})$.

**Hypothesis 1: Conditional entropies decay as a power law with the context length.** The true $n$-gram conditional entropies $H_n$ thus play a fundamental role as a lower bound on the $n$-gram losses $\mathcal{L}_n$. To construct our theory of how $n$-gram losses $\mathcal{L}_n$ change over training, we need to make a hypothesis about how their limiting lower bound $H_n$ varies with context length $n$. Note that conditioning on more data never increases entropy. In essence, the further one looks back in the past (larger $n$), the lower the entropy of the next token (smaller $H_n$). Thus, mathematically, $H_n$ *must* be a non-increasing function of the context length $n$. We make the hypothesis that $H_n$ decreases with $n$ as a power law with exponent $\gamma$, eventually converging to its asymptotic value $H_\infty$:

$$\boxed{H_n - H_\infty \asymp n^{-\gamma}}, \qquad H_\infty = \lim_{n \to \infty} H_n. \tag{21}$$

Here $\asymp$ denotes asymptotic equality to within multiplicative constants, e.g. $f(n) \asymp g(n)$ if and only if there exist constants $C_1, C_2$ and $n_0$ such that $C_1 g(n) \leq f(n) \leq C_2 g(n)$ for all $n \geq n_0$.

**Differential and excess $n$-gram losses.** We want to study how the loss changes as the model becomes capable of using information from context windows of increasing length. Therefore, we define the *differential loss*

$$\Delta_n := \mathbb{E}_{(X_{1:n+1}) \sim p} \left[ -\log \frac{\widehat{p}_i\left(X_{n+1}\Big|X_{1:n}\right)}{\widehat{p}_i\left(X_{n+1}\Big|X_{2:n}\right)} \right]. \tag{22}$$

At each $n$, this differential loss $\Delta_n$ measures the cross-entropy loss when predicting the $n+1$-th token from the previous $n$ tokens minus the cross entropy loss of predicting the $n+1$-th token from the the previous $(n-1)$ tokens. Assuming stationarity of the token-generating process (i.e. that $(X_{1:(n-1)})$ has the same distribution as $(X_{2:n})$, as is usually the case since batches are formed by randomly chunking segments of text), we have that the differential loss is simply

$$\Delta_n = \mathcal{L}_n - \mathcal{L}_{n-1}. \tag{23}$$

Using this relation, we can rewrite the autoregressive loss as a function of the differential losses and the unigram loss:

$$\mathcal{L}_{\mathrm{AR}} = \mathcal{L}_0 + \sum_{n=1}^{T} \frac{T - (n-1)}{T} \Delta_n. \tag{24}$$

Here the unigram loss $\mathcal{L}_0$ is $\mathbb{E}_X[-\log \widehat{p}_0(X)]$. This loss converges quickly over training to the unigram entropy $H_0$, so we will treat it as equal to $H_0$ from now on.

**Hypothesis 2: Drop in differential and excess loss as a function of amount of training data.** Given the role of the differential losses $\Delta_n$ in determining the autoregressive loss $\mathcal{L}_{\mathrm{AR}}$ in (24), to develop our theory of how $\mathcal{L}_{\mathrm{AR}}$ drops with the number of training examples $P$, we make natural assumptions about how the differential losses $\Delta_n = \mathcal{L}_n - \mathcal{L}_{n-1}$ change with $P$. The limit of large $P$ is clear: since $L_n$ converges to $H_n$ in the limit of large $P$, we must have $\Delta_n$ converge to $H_n - H_{n-1}$. Note this is a negative quantity since conditioning on more information typically reduces entropy (e.g. $H_n < H_{n-1}$). Now in the limit of small $P$ and large $n$ there is not enough training data to be able to beneficially use a token $n$ time steps in the past to better predict the next token. So in the limit of small $P$ and large $n$ we expect no additional benefit from the $n$'th token and thus $\mathcal{L}_{n-1} = \mathcal{L}_n$ and therefore $\Delta_n = 0$. However, for each $n$, as $P$ increases, we expect that there will be an $n$ dependent *data threshold* $P_n^*$ at which $\Delta_n$ starts to transition from the $\Delta_n = 0$ data limited regime with $P < P_n^*$ to $\Delta_n = H_n - H_{n-1}$ in the large data regime for $P \gg P_n^*$. We assume that during this transition $\Delta_n(P)$ falls off as a power law with $P$ with exponent $\delta_n$, starting at $\Delta_n(P) = 0$ for $P \leq P_n^*$ and decaying to the asymptotic negative value $\lim_{P \to \infty} \Delta_n(P) = H_n - H_{n-1}$. Thus, we assume the following scaling form,

$$\Delta_n(P) = (H_n - H_{n-1})f_n\left(\frac{P}{P_n^*}\right), \qquad \text{with} \quad \begin{cases} f_n(x) \to 0 & \text{for} \quad x \ll 1, \\ 1 - f_n(x) \to x^{-\delta_n} & \text{for} \quad x \gg 1. \end{cases} \tag{25}$$

Note that, by letting the late power-law-decay exponent depend on $n$, and including edge cases $\delta_n = 0$ for irreducible loss and $\delta_n = \infty$ for faster-than-power-law decay, Eq. 25 is really an assumption about the time- or $n$-dependent data threshold $P_n^*$. Naturally, we expect $P_n^*$, at which the model can start to beneficially use a token $n$ time steps in the past to predict the next token, to increase with $n$. In essence, more data (larger $P_n^*$) is required to successfully predict over longer time horizons (larger $n$). We will next show how to use token-token correlations to estimate $P_n^*$.

But first we define a positive quantity which we call the *excess loss*, which will be useful below in understanding the scaling behavior of the full autoregressive loss $\mathcal{L}_{\mathrm{AR}}$. The excess loss $\mathcal{E}_n(P)$ is defined only for $P \geq P_n^*$ and it is simply the positive height of $\Delta_n(P)$ above its asymptote:

$$\mathcal{E}_n(P) \equiv \Delta_n(P) - (H_n - H_{n-1}) \to -(H_n - H_{n-1})\left(\frac{P_n^*}{P}\right)^{\delta_n} \quad \text{for } P \gg P_n^*. \tag{26}$$

Note this excess loss is positive (or at least non-negative) because $H_n \leq H_{n-1}$, and it decays to 0 as $P \to \infty$.

**Token-token covariances, time-dependent data thresholds, and data-dependent prediction time horizons.** We now use token-token covariances to relate how much data is needed to *start to* successfully predict over what time horizon. We define a two-point token-token covariance matrix as

$$C_{\mu,\nu}(n) = \mathbb{P}\{X_i = \mu, X_{i+n} = \nu\} - \mathbb{P}\{X_i = \mu\}\mathbb{P}\{X_{i+n} = \nu\}, \tag{27}$$

i.e. the covariance of the one-hot representations of two tokens at distance $n$. This is a $v \times v$ correlation matrix where $v$ the vocabulary size. The singular value spectra of such correlation matrices exhibit a small number ($\lesssim 10$) of large singular values, followed by a power-law-decaying bulk of smaller ones. Based on these correlations, we can estimate a time horizon $n$ dependent data threshold $P_n^*$ as follows. Intuitively, we think of $P_n^*$ as the minimum amount of data needed to be able to *start to* successfuly use tokens $n$ time steps in the past to predict the next token (i.e. for $P > P_n^*$ we can start to benefit from tokens $n$ timesteps in the past). A minimal requirement for the beneficial use of tokens $n$ timesteps in the past would be that we could detect the strongest signal in the token-token correlation matrix at time lag $n$, i.e the top singular value of $C_{\mu,\nu}(n)$. The minimal amount of data $P$ required to estimate this top singular value can be obtained via a signal-to-noise argument where the signal is the operator norm of $C(n)$ and the sampling noise is $O(\frac{1}{\sqrt{P}})$. This yields the inequality

$$\|C(n)\|_{\text{op}} > \frac{c}{\sqrt{P}} \Rightarrow P > P_n^* \equiv c^2/\|C(n)\|_{\text{op}}^2. \tag{28}$$

Empirically, we find that the maximal strength of token-token correlations, as measured by the operator norm $\|C(n)\|_{\text{op}}$ decays with time-lag $n$ as a power law with exponent $\beta$, e.g.

$$\|C(n)\|_{\text{op}} \asymp n^{-\beta}. \tag{29}$$

This then yields the time-dependent data threshold

$$P_n^* \asymp n^{2\beta}, \tag{30}$$

which we can use in (25). As expected, to start to successfully predict over longer time horizons $n$, we will need more data (larger $P_n^*$), and the growth of the time-dependent data threshold with $n$ is a power law with exponent $2\beta$.

We can also look at the functional inverse of $P_n^*$ to define

$$n^*(P) \asymp P^{1/2\beta}. \tag{31}$$

We can think of $n^*(P)$ as a data-dependent maximal prediction time horizon. In essence, for a fixed amount of data $P$ we can only beneficially use tokens $n$ time steps in the past to predict the next token if $n < n^*(P)$. As we increase data $P$, the data-dependent maximal prediction time horizon $n^*(P)$ grows with $P$ as a power law with exponent $1/2\beta$. As expected, the more data we have (larger $P$), the further back in the past (larger $n^*(P)$) we can use to predict.

This section concludes how the temporal decay of token-token correlations relate the amount of training data $P$ to the maximal amount of time in the past we can use to successfully predict the next token (e.g. $n^*(P)$).

**Putting it altogether: a theory of the asymptotic decay of the autoregressive loss.** We can now substitute the ansatz for the differential loss $\Delta_n$ in (25) into Eq. 24 for the autoregressive loss $\mathcal{L}_{\text{AR}}$. For a given amount of data $P$, according to our ansatz, $\Delta_n \approx 0$ whenever $P < P_n^*$, or equivalently whenever $n > n^*(P)$. Thus we can restrict the sum from $n = 1, \ldots, T$ in Eq. 24 to $n = 1, \ldots, n^*(P)$. In essence for a given $P$, differential losses $\Delta_n$ for $n$ beyond the data-dependent maximal prediction time horizon $n^*(P)$ do not contribute to $\mathcal{L}_{\text{AR}}$. Thus we obtain

$$\mathcal{L}_{\text{AR}} = H_0 + \sum_{n=1}^{n^*(P)} \frac{T - (n-1)}{T} (H_n - H_{n-1}) + \sum_{n=1}^{n^*(P)} \frac{T - (n-1)}{T} \mathcal{E}_n(P), \tag{32}$$

where we have used our expression for the excess loss $\mathcal{E}_n(P)$ in Eq. 26. We can further drop the $O(1)$ coefficients $\frac{T-(n-1)}{T}$ to obtain the asymptotic equality up to constants,

$$\mathcal{L}_{\text{AR}}(P) \asymp H_0 + \sum_{n=1}^{n^*(P)} (H_n - H_{n-1}) + \sum_{n=1}^{n^*(P)} \mathcal{E}_n(P) \tag{33}$$

Collapsing the sum over entropies finally yields the appealing expression

$$\mathcal{L}_{\text{AR}}(P) \asymp H_{n^*(P)} + \sum_{n=1}^{n^*(P)} \mathcal{E}_n(P). \tag{34}$$

This expression reveals two qualitatively distinct contributions to the autoregressive loss. The first term is a "boundary term" corresponding to the conditional entropy of the next token given all tokens within the data-dependent maximal prediction time horizon $n^*(P)$ and it is a lower bound on $\mathcal{L}_{n^*(P)}$. For a given $P$, under our theory, we assume the model cannot use past times $n > n^*(P)$ to predict the next token, so this conditional entropy reflects the contribution to the loss due to this finite prediction time horizon. However, the model may also make *suboptimal* use of the tokens at past times $n \le n^*(P)$ *within* the prediction time horizon. This suboptimal use of tokens within the time horizon is captured by the excess loss, and yields the sum of (positive) excess losses in the second term in Eq. 34. Thus as $P$ increases, the model can learn (reduce

$\mathcal{L}_{AR}(P)$) in two qualitatiatively distinct ways: (1) successfully predict over longer time horizons, reducing the first term; and (2) make better use of information within the shorter time horizons it had access to even with less data, reducing the second term. Clearly the first option for reducing the loss is only available if the data-dependent maximal prediction time horizon $n^*(P)$ is much less than the maximal context length $T$. Therefore we will be interested in this regime, otherwise the maximal context length $T$ would limit loss reduction in a manner not accounted for by our theory.

We can now insert our power law ansatzes to estimate the strengths of the first boundary term due to a finite prediction time horizon, and the second term measuring excess loss due to suboptimal use of information within the prediction time horizon. We expect power law behavior for the difference between $\mathcal{L}_{AR}(P)$ and its large $P$ asymptotic value $H_\infty$:

$$\mathcal{L}_{AR}(P) - H_\infty \asymp \left[ H_{n^*(P)} - H_\infty \right] + \sum_{n=1}^{n^*(P)} \mathcal{E}_n(P). \tag{35}$$

The scaling behavior for the first term is straight forward: we have $n^*(P) \asymp P^{1/2\beta}$ from Eq. 31, and $H_n - H_\infty \asymp n^{-\gamma}$ from Eq. 21. Putting these together we obtain,

$$H_{n^*(P)} - H_\infty \asymp P^{-\frac{\gamma}{2\beta}}. \tag{36}$$

This scaling reflects how part of the loss decreases with increasing $P$ specifically due to an increase in the data-dependent prediction time horizon $n^*(P)$.

Next, for the scaling behavior of the second term involving the sum of excess losses, in the explicit expression for the excess loss $\mathcal{E}_n(P)$ in Eq. 26, we will set $\delta_n = \delta$ for all $n$'s for simplicity (in general we can replace $\delta$ with $\min_n \delta_n$). Now inserting the ansatzes $H_n \asymp n^{-\gamma} \implies -(H_n - H_{n-1}) \asymp n^{-\gamma-1}$ and $P_n^* \asymp n^{2\beta}$ from Eq. 30 into Eq. 26, we obtain

$$\mathcal{E}_n(P) \asymp n^{-\gamma-1} \left( \frac{n^{2\beta}}{P} \right)^\delta. \tag{37}$$

Note the larger $\delta$ is, the quicker the excess loss falls off with $P$, and therefore the quicker the model can learn to optimally use the information in the tokens within its prediction time horizon as the amount of data $P$ increases. Now we must sum $\mathcal{E}_n(P)$ from $n = 1$ to the data-dependent time horizon $n^*(P) = P^{\frac{1}{2\beta}}$ obtaining

$$\sum_{n=1}^{n^*(P)} \mathcal{E}_n(P) \asymp \frac{1}{P^\delta} \sum_{n=1}^{P^{\frac{1}{2\beta}}} n^{2\beta\delta-\gamma-1}. \tag{38}$$

Now approximating the sum with an integral we obtain

$$\sum_{n=1}^{n^*(P)} \mathcal{E}_n(P) \asymp \frac{1}{P^\delta} \int_1^{P^{\frac{1}{2\beta}}} s^{2\beta\delta-(\gamma+1)} \, ds \asymp \frac{1}{P^\delta} \begin{cases} \text{const.} & \text{if} \quad \frac{\gamma}{2\beta} > \delta, \\ \log P & \text{if} \quad \frac{\gamma}{2\beta} = \delta, \\ P^{\delta-\frac{\gamma}{2\beta}} & \text{if} \quad \frac{\gamma}{2\beta} < \delta. \end{cases} \tag{39}$$

This yields 2 qualitatively distinct regimes. First if $\frac{\gamma}{2\beta} < \delta$ corresponding to rapid learning within the prediction time horizon, the second term in Eq. 35 decays with $P$ at the same rate as the first term, namely $P^{-\frac{\gamma}{2\beta}}$, and therefore the entire autoregressive loss decays with $P$ in this manner. However, if $\frac{\gamma}{2\beta} > \delta$, corresponding to slow learning within the prediction time horizon, the second term in Eq. 35 decays as $P^{-\delta}$ which is a slower decay than the first term, which is still $P^{-\frac{\gamma}{2\beta}}$. In this slow learning regime, the model is racing ahead and increasingly using longer and longer prediction time horizons as $P$ increases, but it is slowly learning how to use the information within the time horizon well for prediction. In this case the smaller exponent $\delta$ of the second term dominates the autoregressive loss, which then decays as $P^{-\delta}$.

Overall, putting all of this altogether, we obtain our final theoretical prediction for the scaling of the autoregressive loss with data:

$$\boxed{\mathcal{L}_{AR}(P) - H_\infty \asymp P^{-\min\left\{\delta, \frac{\gamma}{2\beta}\right\}}.} \tag{40}$$

Intriguingly, in the within time horizon the fast learning regime $\delta > \frac{\gamma}{2\beta}$ (in which information within the prediction time horizon is quickly learned well before the time horizon extends as $P$ increases), the decay exponent is simply $\frac{\gamma}{2\beta}$ which *only depends on two statistical properties of natural language itself*, namely the decay of $n$-gram entropy with $n$, governed by $\gamma$, and the decay of correlations with $n$ governed by $\beta$. This allows, to our knowledge for the first time, to measure statistical properties of natural language to predict the exponent of a neural scaling law for language models, assuming they are operating in the fast learning regime.

**A theory of collapse of $n$-gram loss learning curves.** A simpler and more general prediction of our theory is as follows. We can decompose the full autoregressive learning curve $\mathcal{L}_{AR}(P)$ into individual $n$-gram loss learning curves $\mathcal{L}_n(P)$, as seen above. When plotting each of these losses as a function of data amount $P$, these curves will generally be distinct and not lie on top of each other in the loss ($\mathcal{L}$) versus data ($P$) plane.

However, under our theory, each curve $\mathcal{L}_n(P)$ has its own time horizon or $n$-dependent data threshold $P^*(n)$. As discussed above, this is the minimal data amount at which the model can start to use tokens at time horizon $n$ to predict the next token. Therefore, when plotting $\mathcal{L}_n(P)$, it can be useful to plot it as a function of the data amount $P$ in units of the data threshold $P^*(n)$, i.e., in terms of the rescaled data amount $P/P^*(n)$.

Similarly, the vertical axis of loss for $\mathcal{L}_n$ also has a natural scale, namely $H_n$, which, as discussed above, is the value to which $\mathcal{L}_n(P)$ asymptotes as $P \to \infty$. Therefore, when plotting $\mathcal{L}_n$, it can be useful to plot it in units of $H_n$, i.e. the rescaled value $\mathcal{L}_n/H_n$.

This suggests the curves

$$\ell(P/P_n^*) \equiv \frac{\mathcal{L}_n(P)}{H_n} \tag{41}$$

may be more similar to each other, and could even collapse on top of each other. Now inserting our ansatz $H_n \asymp n^{-\gamma}$ and $P_n^* \asymp n^{2\beta}$, we find

$$\ell(P/n^{2\beta}) \equiv n^\gamma \mathcal{L}_n(P). \tag{42}$$

The scaling collapse of $\mathcal{L}_n$ can actually be derived from our ansatz on the differential loss Eq. 25, under the additional assumption that the variations of $\delta_n$ are negligible for large $n$ so that $\delta_n = \delta$ and $f_n = f$.

$$\mathcal{L}_n = H_0 + \sum_{1 \leq n' \leq n} (H_{n'} - H_{n'-1}) f\left(\frac{P}{P_{n'}^*}\right)$$

$$\implies \mathcal{L}_n = H_n + \sum_{1 \leq n' \leq n} (H_{n'} - H_{n'-1})\left(f\left(\frac{P}{P_{n'}^*}\right) - 1\right)$$

$$\implies \mathcal{L}_n \asymp H_\infty + C_0 n^{-\gamma} + \int_{1 \leq n' \leq n} C_1 n^{-\gamma-1}\left(f\left(\frac{P}{(n')^{2\beta}}\right) - 1\right) dn$$

$$\implies \mathcal{L}_n - H_\infty \asymp C_0 n^{-\gamma} + C_1 \gamma n^{-\gamma} \int_{0 \leq u \leq 1} u^{-(1+\gamma)}\left(f\left(\frac{P}{n^{2\beta}u^{2\beta}}\right) - 1\right) du$$

$$\implies \mathcal{L}_n - H_\infty \asymp n^{-\gamma} \ell\left(\frac{P}{n^{2\beta}}\right), \qquad \text{with } \ell(x) \sim \text{const.} + x^{-\delta} \qquad \text{for } x \gg 1. \tag{43}$$

Thus, our scaling theory provides a striking prediction: we simply measure two properties of language, the entropy exponent $\gamma$ and the correlation exponent $\beta$, and we plot the rescaled versions of all the disparate $n$-gram losses in Eq. 42, then they should all collapse onto the same curve. Remarkably, this is what we see in many cases in the main paper, thereby providing strong evidence for our theory.

## B. Sampling noise in empirical token-token correlations

In this appendix, we justify the $P^{-1/2}$ sampling-noise scale used in the signal-to-noise argument leading to Eq. 4. Fix a time lag $n$. Given $P$ example pairs of tokens $(X_i, X_{i+n})$ sampled independently from the corpus, we estimate the lag-$n$ joint probabilities by empirical averages

$$\widehat{p}_{\mu\nu} = \frac{1}{P} \sum_{i=1}^{P} \mathbf{1}\{X_i = \mu, \ X_{i+n} = \nu\}, \tag{44}$$

and the corresponding marginals by

$$\widehat{p}_{\mu} = \frac{1}{P} \sum_{i=1}^{P-n} \mathbf{1}\{X_i = \mu\}, \qquad \widehat{p}'_{\nu} = \frac{1}{P} \sum_{i=1}^{P} \mathbf{1}\{X_{i+n} = \nu\}. \tag{45}$$

The empirical lag-$n$ covariance matrix is then

$$(\widehat{C}_P(n))_{\mu,\nu} = \widehat{p}_{\mu\nu} - \widehat{p}_{\mu}\widehat{p}'_{\nu}, \tag{46}$$

which estimates the population covariance

$$C_{\mu\nu}(n) = \mathbb{P}\{X_i = \mu, X_{i+n} = \nu\} - \mathbb{P}\{X_i = \mu\}\mathbb{P}\{X_{i+n} = \nu\}. \tag{47}$$

For fixed vocabulary size and fixed lag $n$, each entry of $\widehat{C}_P(n)$ is built from empirical averages of bounded random variables, which obey central-limit-theorem scaling. Thus, for every fixed pair $(\mu, \nu)$, with high probability over the sampling of the $P$ token pairs,

$$\widehat{C}_P(n)_{\mu\nu} - C(n)_{\mu\nu} = O(P^{-1/2}). \tag{48}$$

Equivalently,

$$\widehat{C}_P(n) = C(n) + \Xi_P(n), \tag{49}$$

where $\Xi_P(n)$ is an additive sampling-noise matrix whose entries have typical size $P^{-1/2}$.

The same scaling also holds for the operator norm, up to constants that depend on the vocabulary size. One way to make this statement rigorous is to write the empirical covariance estimator as a sum of centered and bounded single-sample random matrices built from the indicators $\mathbf{1}\{X_i = \mu, \ X_{i+n} = \nu\}$. Then, the matrix Bernstein inequality gives, with high probability for large $P$,

$$\|\Xi_P(n)\|_{\mathrm{op}} \lesssim \sqrt{\frac{\sigma_n^2 \log V}{P}}, \tag{50}$$

where $V$ is the vocabulary size and $\sigma_n^2$ bounds the variance of the single-sample random matrix.

Denote the singular values of $C(n)$ and $\widehat{C}_P(n)$ by

$$\sigma_1(n) \geq \sigma_2(n) \geq \cdots, \qquad \widehat{\sigma}_1(n) \geq \widehat{\sigma}_2(n) \geq \cdots. \tag{51}$$

By Weyl's inequality for singular values,

$$|\widehat{\sigma}_k(n) - \sigma_k(n)| \leq \|\widehat{C}_P(n) - C(n)\|_{\mathrm{op}} = O\left(\sqrt{\frac{1}{P}}\right) \tag{52}$$

for every $k$. Thus, a singular mode of strength $\sigma_j(n)$ is spectrally resolvable only if it lies above the sampling-noise floor, i.e.

$$\sigma_k(n) \gtrsim O\left(\sqrt{\frac{1}{P}}\right). \tag{53}$$

Consequently, the strongest lag-$n$ correlation mode $\sigma_1(n)$ is detectable only when its signal, measured by $\|C(n)\|_{\mathrm{op}} \asymp n^{-\beta}$, is larger than the sampling-noise floor. Equivalently, the data threshold for resolving correlations at lag $n$ satisfies

$$P_n^* \asymp \|C(n)\|_{\mathrm{op}}^{-2} \asymp n^{2\beta}. \tag{54}$$

Inverting this relation gives the data-dependent prediction horizon

$$n^*(P) \asymp P^{1/(2\beta)}, \tag{55}$$

as defined in the main text. An alternative way to formulate the same signal-to-noise criterion is in terms of the Frobenius norm rather than the operator norm. This corresponds to asking when the total lag-$n$ correlation energy $\sum_k \sigma_k^2$, rather than only the strongest singular mode, is distinguishable from sampling noise. Since Frobenius and operator norms display the same scaling with $n$, the two formulations yield the same scaling prediction.

The signal-to-noise criterion on the operator norm is also analogous to the spectral detectability threshold in spiked random-matrix models (Baik et al., 2005): a low-rank population signal becomes visible in the leading empirical singular mode only once it separates from the sampling-noise bulk.

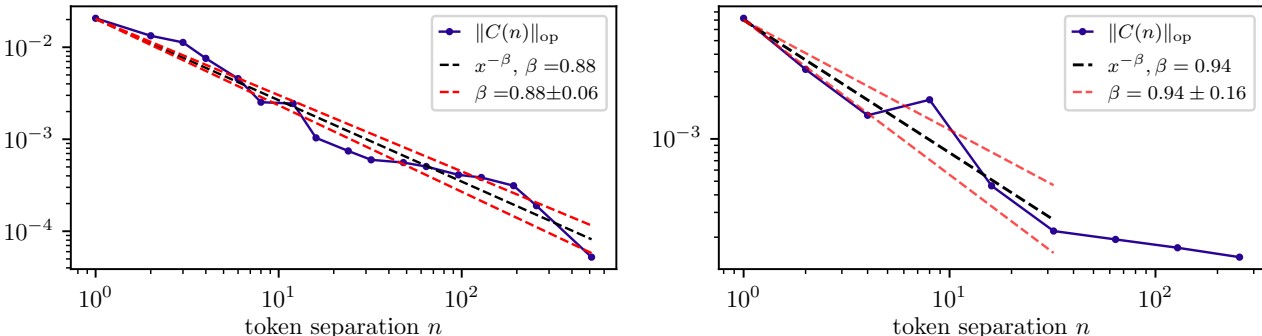

*Figure 7.* Comparison of the temporal decay of the correlations operator norm (purple solid) with its power-law fit $n^{-\beta}$ (black dashed) and power laws $n^{-(\beta\pm\Delta_\beta)}$ (red dashed) with $\Delta_\beta$ the standard error reported in Eq. 13 (TinyStories, left panel) and Eq. 14 (WikiText, right panel).

## C. Errors

In this appendix we describe how we estimate the uncertainties reported in the main text for the language exponents $\gamma$ and $\beta$, how these uncertainties propagate to the predicted data-limited scaling exponent $\alpha_D = \gamma/(2\beta)$, and how robust the observed scaling collapses and empirical learning-curve fits are to variations within these uncertainty ranges.

### C.1. Errors on $\beta$ and $\gamma$

**Correlations exponent $\beta$.** The exponent $\beta$ is obtained as the slope of a linear regression fit between $m$ distinct $(\log n, \log \|C(n)\|_{\mathrm{op}})$ pairs. We can estimate the error $\Delta_\beta$ as the standard error on the fitted slope. An additional estimate comes from bootstrapping: resample the $m$ pairs with replacement from the original set $N_{\mathrm{b}}$ times, compute the slope for each bootstrap sample, them take the stantard deviation over bootstrap samples. For TinyStories we take 15 logarithmically equidistant pairs between $n = 1$ and $n = 200$, since most stories end within 200 tokens. Standard error and bootstrapping error are comparable and equal to 0.06 within the chosen precision. For WikiText, as explained in the main text, we focus on the small-$n$ decay, taking 6 logarithmically equidistant pairs between $n = 1$ and $n = 32$. The standard error of the fitted slope, 0.13, is smaller than the bootstrap estimate, 0.16, reported in Eq. 14.

**Entropy exponent $\gamma$.** For each model class considered, we can estimate $\gamma$ from a power-law fit of the portion of the $\mathcal{L}_n(P)$-$n$ curve which has approached its $P \to \infty$ value. For TinyStories, we estimate $\gamma$ from the learning curve of the GPT-2-style transformer with maximal context length $T = 128$. We use this model class and context length because this is the setting in which we performed the most thorough hyperparameter tuning, and where the convergence of the small-$n$ losses to a stable limiting curve is clearest. Eq. 11 reports the exponent of the power-law fit of the limiting curve over the range $n = 1, \ldots, 16$ and the corresponding bootstrapping error. To check that this estimate is not an artefact of this particular architecture or context length, we repeat the same fit over the same range of $n$ for the other model classes considered in the paper. The resulting exponents are compatible with the GPT-2, $T = 128$ estimate, in the sense that their $95\%$ confidence intervals, computed from the bootstrap standard errors, overlap with that of the reference fit. For WikiText, the estimation of $\gamma$ is more delicate because the available dataset sizes only allow the small-$n$ part of the $L_n$ curve to approach its limiting value. In practice, only the first few points show clear convergence across training-set sizes. We therefore use the first three points to obtain the central estimate of $\gamma$, since these are the points for which convergence is most reliable. However, a bootstrap estimate of the uncertainty from only three points is essentially degenerate, because the log-log fit is almost fully constrained. To obtain a more conservative finite-sample error estimate, we compute the bootstrap standard error using the first four points instead. This yields the value reported in Eq. 12. This error bar should therefore be interpreted as a finite-range uncertainty associated with the choice of fitting window, rather than as a purely statistical standard error for a fixed asymptotic regime.

### C.2. Quality of collapses within the error range

We next ask how sensitive the scaling collapses are to variations of the exponents $\gamma$ and $\beta$. For $\beta$, we use the uncertainty range obtained directly from the bootstrap standard error of the correlation decay fit. Thus, for each dataset, we compare the

collapse obtained with the central value of $\beta$ to the collapses obtained at $\beta \pm \epsilon_\beta$.

For $\gamma$, we use a slightly broader and more conservative range. The bootstrap error for a fixed model class and fitting window can be smaller than the systematic variation obtained by repeating the entropy-decay fit across different model classes. Therefore, for TinyStories we define the displayed range of $\gamma$ as the interval from the minimum lower endpoint to the maximum upper endpoint of the 95% confidence intervals obtained from all model classes used to estimate $\gamma$, always fitting over the same range of horizons. This gives a range that captures both the statistical uncertainty of each fit and the architecture-dependent variation in the limiting $L_n$ curve. We then visualize the collapse at the lower endpoint, central value, and upper endpoint of this range in Fig. 9. For TinyStories, the collapse is stable throughout this range of $\gamma$ and deteriorates visibly when $\beta$ is moved outside its bootstrap uncertainty range (Fig. 8). This supports the claim that the independently measured exponents are also the values that organize the $n$-gram learning curves. For WikiText, within the bootstrap uncertainty range of $\gamma$ (Eq. 12), the collapse doesn't qualitatively change. As a sensitivity check, we therefore vary $\gamma$ beyond this range in Fig. 12, choosing a range wide enough to show clear deterioration of the collapse. For $\beta$, as in TinyStories, the collapse deteriorates visibly when $\beta$ is moved outside its bootstrap uncertainty range (Fig. 11).

There is one exception to this qualitative picture. For the LLaMA model trained on TinyStories, the sharpest visual collapse appears to occur for a smaller value, approximately $\beta \simeq 0.65$, which lies below the independently measured range $\beta = 0.88 \pm 0.06$. Since the corresponding autoregressive scaling exponent remains compatible with our final prediction $\alpha_D = \gamma/(2\beta)$, this departure does not affect the main exponent-level conclusion of the paper. We therefore treat it as a finite-range or architecture-dependent feature of the collapse and leave its detailed interpretation to future work.

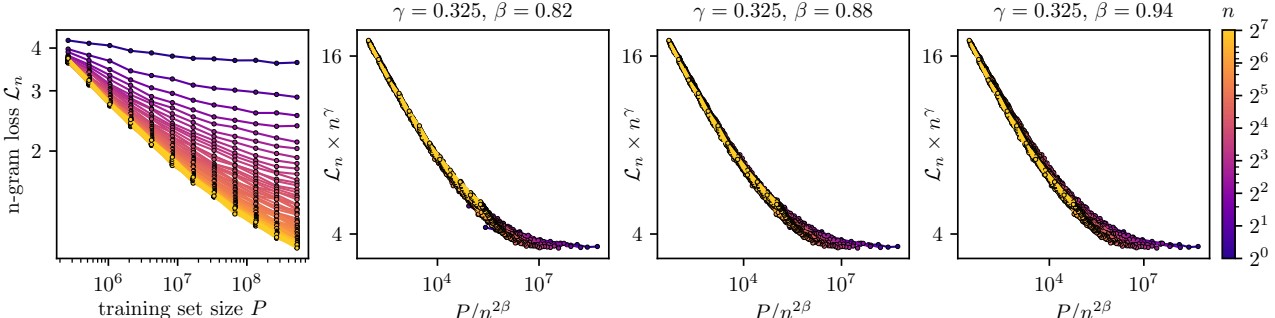

*Figure 8.* Qualitative deterioration of collapse for TinyStories while leaving $\gamma$ fixed and sweeping $\beta$ within the uncertainty range in Eq. 13.

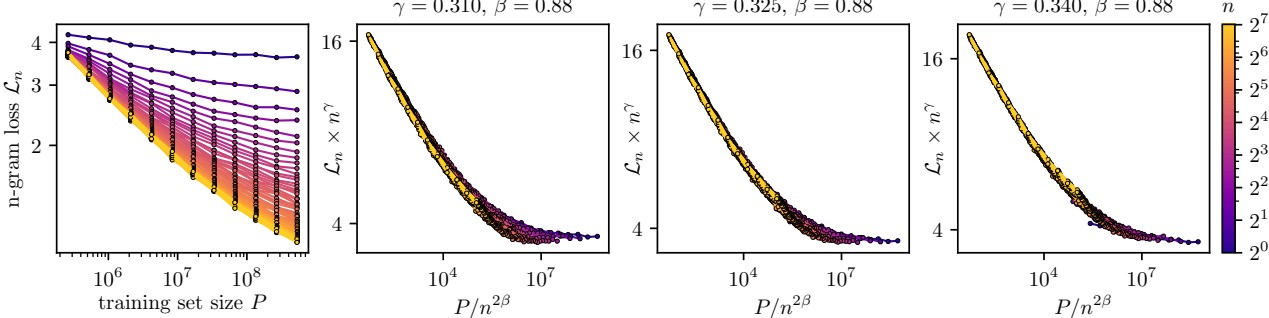

*Figure 9.* Qualitative deterioration of collapse for TinyStories while leaving $\beta$ fixed and sweeping $\gamma$ past the uncertainty range in Eq. 11.

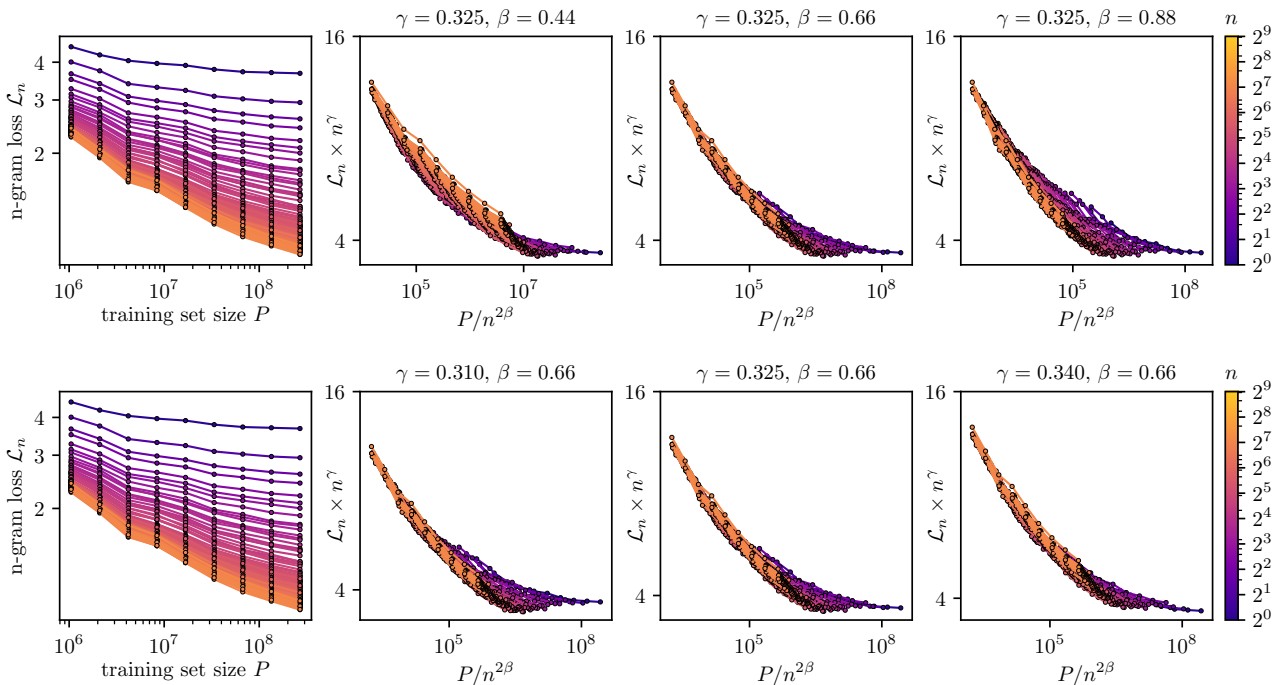

*Figure 10.* For the learning curves of Llama, the collapse appears sharper for $\beta \in [0.6, 0.7]$, outside of the uncertainty range in Eq. 13.

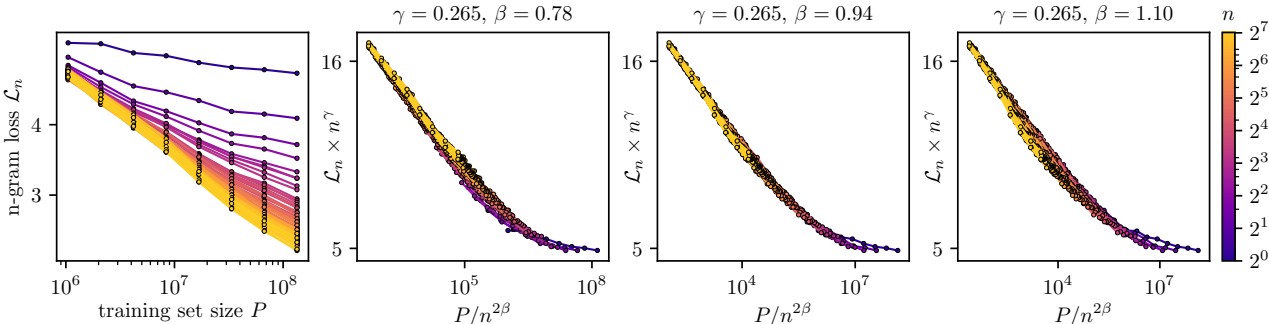

*Figure 11.* Qualitative deterioration of collapse for WikiText while leaving $\gamma$ fixed and sweeping $\beta$ within the uncertainty range in Eq. 14.

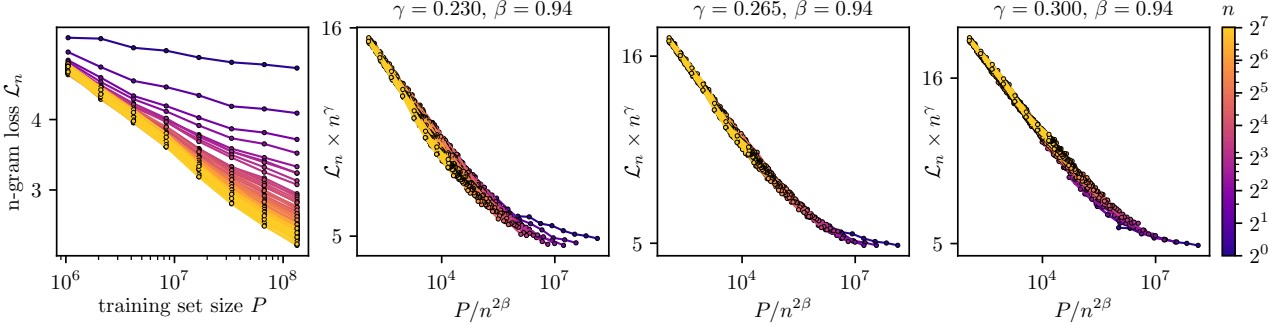

*Figure 12.* Qualitative deterioration of collapse for WikiText while leaving $\beta$ fixed and sweeping $\gamma$ past the uncertainty range in Eq. 12.

### C.3. Error on the data-limited exponent and fits of the empirical learning curves

To estimate the error on our prediction of $\alpha_D$ from the uncertainties on $\beta$ and $\gamma$ we employ a standard propagation-of-error formula,

$$\Delta_\alpha = \sqrt{\left(\frac{1}{2\beta}\right)^2 \Delta_\gamma^2 + \left(\frac{\gamma}{2\beta^2}\right)^2 \Delta_\beta^2}, \tag{56}$$

which yields the ranges reported in the main text. To compare this prediction with the empirical learning curves, we estimate $\hat{\alpha}_D$ directly from the lower envelope of the autoregressive losses obtained from the same model class at different context lengths $T$. For each value of $m \geq 5$, we fit the first $m$ points of this lower envelope on the log-log scale. Since the empirical scaling has the form $P^{-\hat{\alpha}_D}$, the fitted slope is negative and we report the corresponding positive exponent $\hat{\alpha}_D$. The table below reports the $95\%$ bootstrap confidence intervals obtained in this way, together with the interval predicted from the independently measured language statistics.

**Comparison of predicted and empirical data-limited exponents**

|            | TinyStories        | WikiText           |
|------------|--------------------|--------------------|
| Prediction | [0.172, 0.198]     | [0.116, 0.166]     |
| $m = 5$    | [0.1770, 0.1946]   | [0.1324, 0.1552]   |
| $m = 6$    | [0.1786, 0.1893]   | [0.1341, 0.1685]   |
| $m = 7$    | [0.1786, 0.1869]   | [0.1374, 0.1673]   |
| $m = 8$    | [0.1730, 0.1851]   | [0.1467, 0.1605]   |
| $m = 9$    | [0.1689, 0.1838]   |                    |
| $m = 10$   | [0.1633, 0.1807]   |                    |
| $m = 11$   | [0.1566, 0.1782]   |                    |
| $m = 12$   | [0.1494, 0.1751]   |                    |

# D. Supporting figures

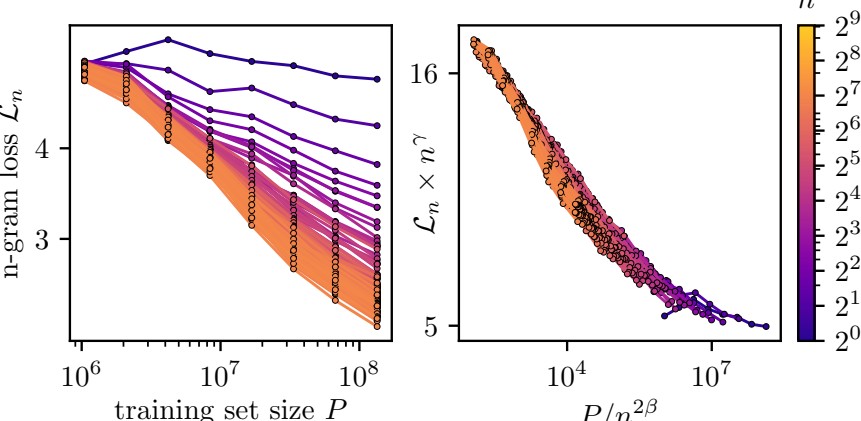

*Figure 13.* $n$-**gram loss collapse for GPT-2-style transformers trained on WikiText at** $T = 512$. (Same as Fig. 4 (Top), but for $T = 512$, rather than $T = 128$.) Note that Fig. 4 (Bottom) contains the auto-regressive loss $\mathcal{L}(P)$ for both $T = 128$ and $T = 512$, so we do not duplicate it here.

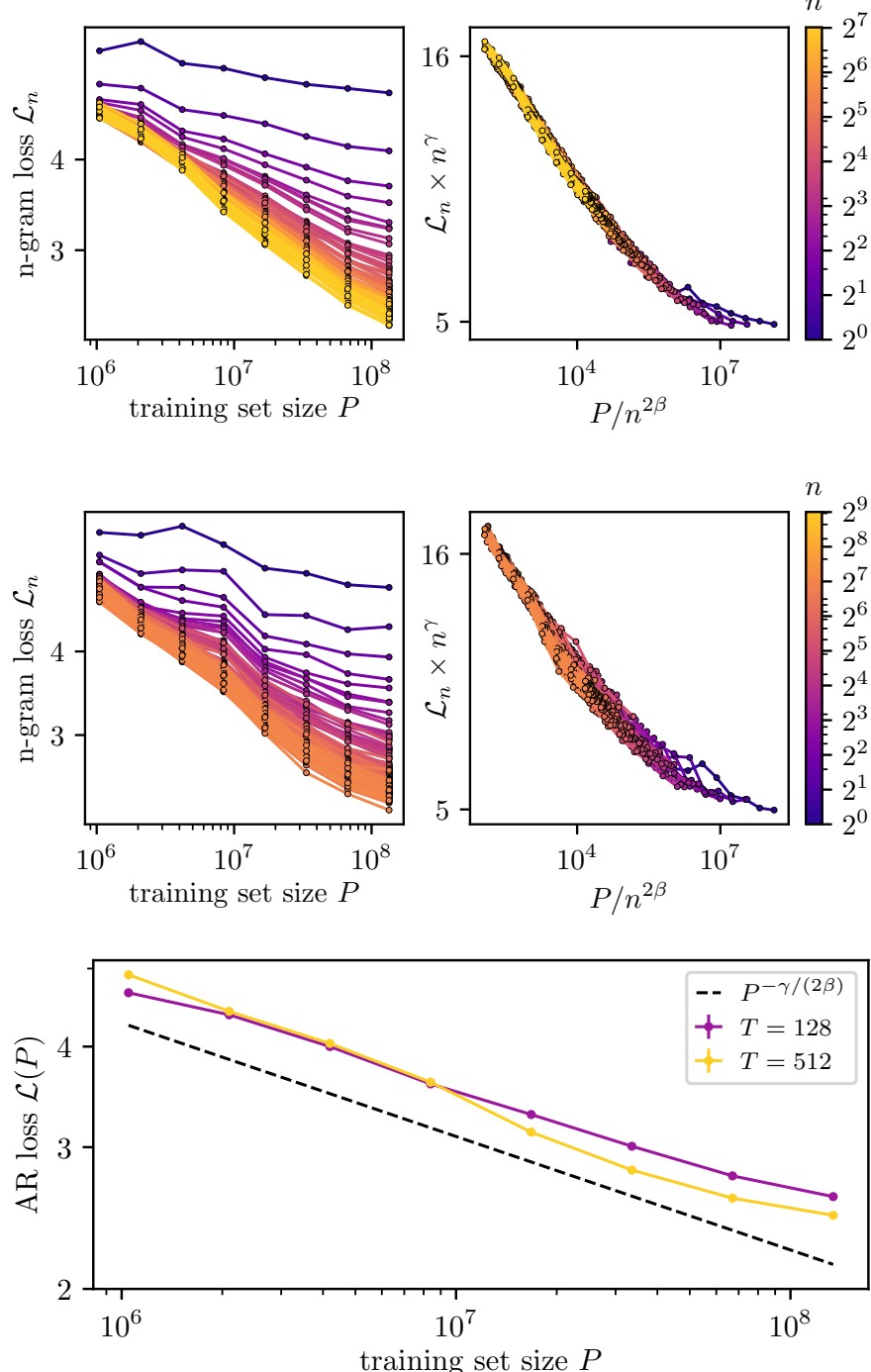

*Figure 14.* $n$**-gram collapse and data-limited scaling exponent prediction for GPT-2-style transformers with RoPE trained on WikiText.** (Same as Fig. 4, but uses RoPE.) **Top** is for $T = 128$, **Middle** for $T = 512$, and Bottom displays the scaling laws of both cases.

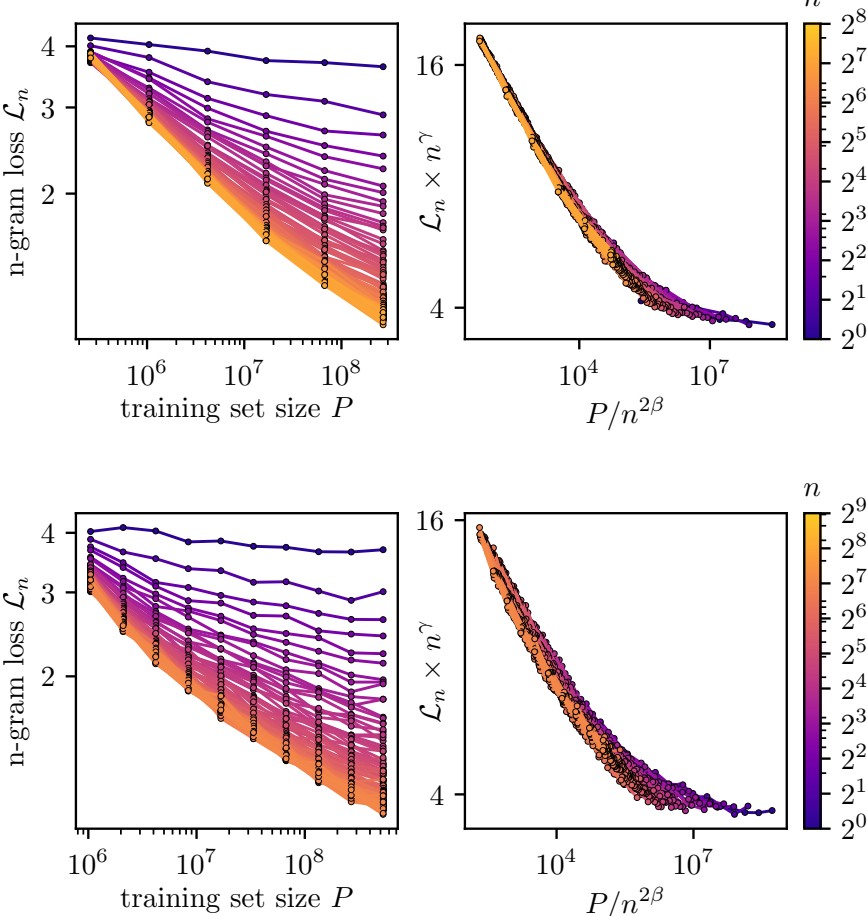

*Figure 15.* $n$**-gram loss collapse for GPT-2-style transformers trained on TinyStories** at $T = 64$, $256$ **and** $512$. (Same as Fig. 1 (Top), but for different $T$.) Note that Fig. 1 (Bottom) contains the auto-regressive loss $\mathcal{L}(P)$ for all $T$'s, so we do not duplicate it here.

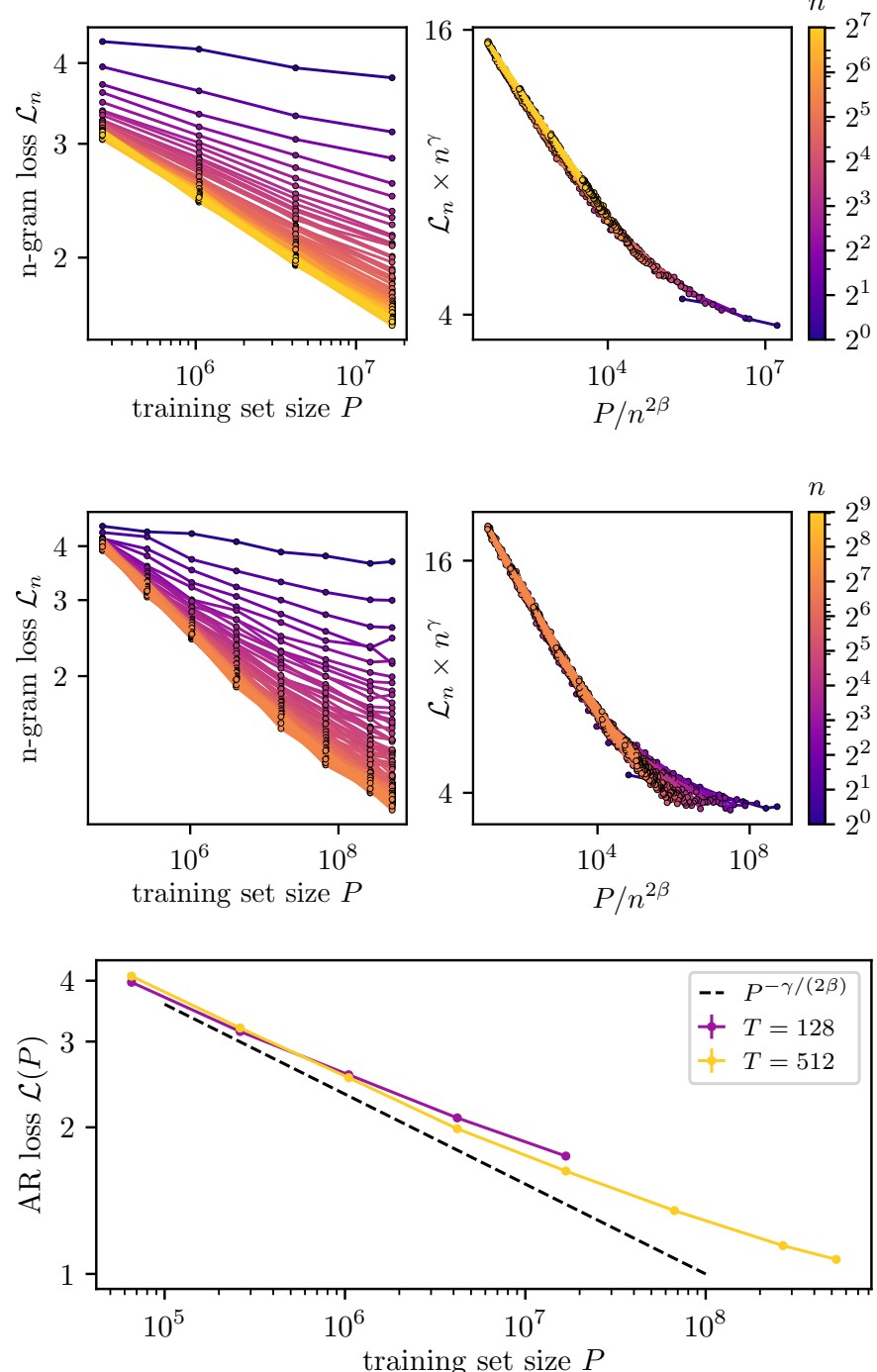

*Figure 16.* $n$**-gram collapse and data-limited scaling exponent prediction for GPT-2-style transformers with RoPE trained on TinyStories.** (Same as Fig. 1, but uses RoPE.) The scaling collapse on the top row refers to $T = 128$, the one on the middle row to $T = 512$.

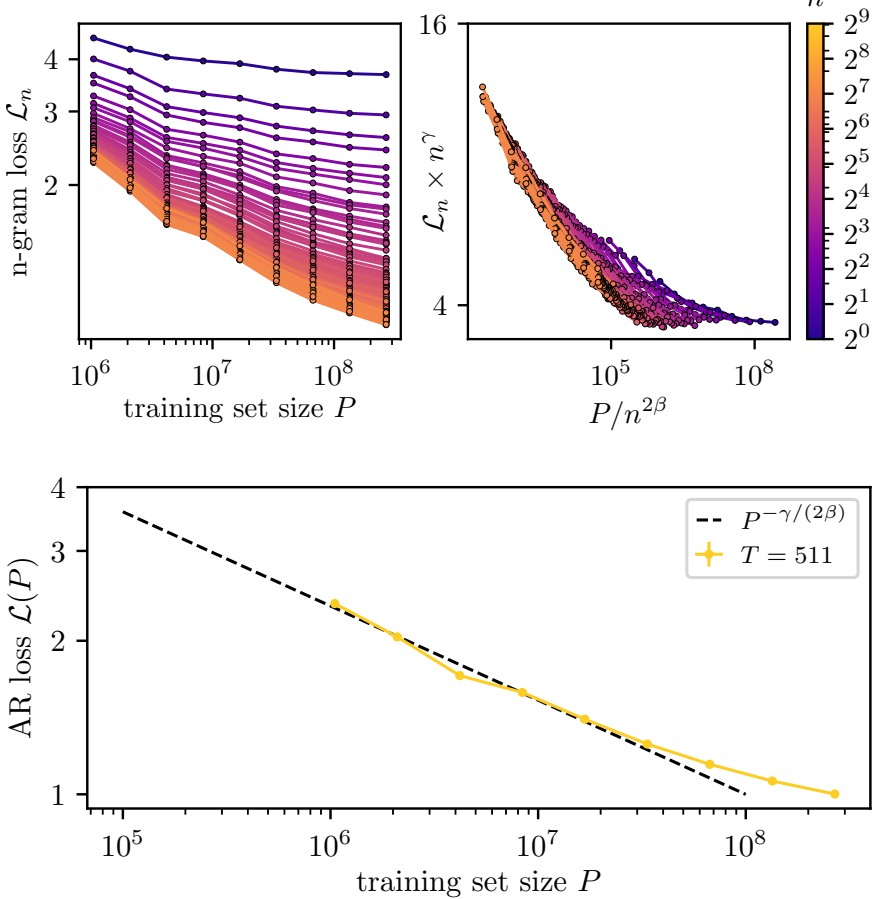

*Figure 17.* $n$-**gram collapse and data-limited scaling exponent prediction for a Llama-style transformer trained on TinyStories.** (Same as Fig. 1, but uses Llama). Having $\beta$ smaller than 0.88 achieves a better collapse, as detailed in Fig. 10. Nonetheless, the scaling law is compatible with $\alpha_D = \gamma/(2\beta)$ with $\gamma = 0.325$ and $\beta = 0.88$.

# E. Experiment details

In this section, we outline several key design decisions made for our experiments. The code used to train all language models considered in this paper, including the data and scripts necessary to reproduce all the figures, is available at `https://github.com/fracagnetta/small-language-modelling`.

**Datasets and tokenization.** Our experiments use the TinyStories (Ronen & Yuanzhi, 2023) and WikiText (Merity et al., 2017) (wikitext-103-raw-v1) datasets. All text is tokenized using a two-stage procedure consisting of whitespace pre-tokenization followed by byte-pair encoding (BPE) (Gage, 1994; Sennrich et al., 2016) with a vocabulary size of 8192. The resulting token sequences are concatenated into a single stream, with an end-of-sequence (EOS) token inserted between documents. For TinyStories, we report losses on the validation set, while for WikiText we concatenate the validation and test sets for evaluation, as each split is small.

**Architectures.** Most experiments use a GPT-2–style architecture (Radford et al., 2019), implemented based on the nanoGPT codebase (Karpathy, 2022). The base configuration uses an embedding dimension of 768, 12 layers, 12 attention heads, and a feedforward dimension equal to four times the hidden size (denoted $(d_{\text{emb}} = 768, d = 12, n_h = 12, \text{ffwd} = 4)$), for a total of approximately 98M trainable parameters. For TinyStories experiments, we find this base architecture sufficient, and only consider slight variations in embedding dimension and depth to verify that larger models do not achieve lower validation loss. For WikiText experiments, we additionally consider two larger model sizes: $(d_{\text{emb}} = 1152, d = 16, n_h = 12, \text{ffwd} = 4)$ with 274M parameters, and $(d_{\text{emb}} = 1728, d = 16, n_h = 16, \text{ffwd} = 4)$ with 600M parameters. For each dataset size, we check whether increasing model size leads to improvements in validation loss. All GPT2-style model experiments are performed both with absolute positional embeddings and with rotary positional embeddings.

For LLaMA experiments, we use a reduced version of the LLaMA-3.2-1B architecture (Grattafiori et al., 2024) implemented using the HuggingFace Transformers library (Wolf et al., 2020), with the vocabulary size set to 8192 and tied input and output embeddings. We consider three model sizes, denoted $(d_{\text{emb}}, d, n_h, n_{kv}, d_{ff})$: $(768, 12, 12, 12, 2688)$, $(1024, 16, 16, 8, 3584)$, and $(2048, 16, 16, 8, 7168)$. The intermediate model size is found to be sufficient for all dataset sizes considered.

For Mamba experiments, we use a stack of $d$ standard Mamba blocks from the mamba-ssm package at `https://github.com/state-spaces/mamba`. The base configuration uses 12 layers and an embedding dimension of 768 (as in the GPT-2 setup), with the depthwise convolution kernel size set to the default 4 and the inner-dimension expansion factor set to the default 2.

**Training hyperparameters.** For each model architecture and dataset size $P$, we tune optimization hyperparameters to achieve the lowest possible validation loss. We use the AdamW optimizer (Kingma & Ba, 2017; Loshchilov & Hutter, 2019) and perform grid search over the learning rate, weight decay, number of training epochs, and batch size.

For experiments using the GPT-2 architecture (including both absolute positional embeddings and RoPE) on both datasets, we search over the following grid:

- Learning rate: (3e-4, 1e-3, 3e-3)

- Weight decay: (0, 3e-3, 1e-2, 3e-2)

- Number of training epochs: (8, 12, 14, 16, 20)

- Batch size: (1, 2, 4, 8, 16)

Mamba experiments on TinyStories use the same grid.

For LLaMA experiments on TinyStories, the batch size is fixed to 64, and we tune the learning rate, weight decay, and number of training epochs following (Kim et al., 2025). In this setting, we search over:

- Learning rate: (3e-4, 1e-3, 3e-3)

- Weight decay: (0.3, 1, 3, 10)

- Number of training epochs: (8, 12, 16, 20, 24, 28, 32, 64)

Due to computational constraints, we do not perform exhaustive hyperparameter search, particularly at the largest dataset sizes. Instead, following (Kim et al., 2025), we aim for local optimality wherever feasible, in the sense that increasing or decreasing any single hyperparameter does not improve validation loss. In practice, we find that the optimal hyperparameters at a given dataset size provide a good initialization for tuning at the next larger dataset size.

**Compute.** All experiments are implemented in PyTorch (Paszke et al., 2019), and LLaMA experiments are trained using the HuggingFace Transformers library (Wolf et al., 2020). All experiments are run on NVIDIA H100 GPUs. GPT-2 experiments use a single GPU, with runtimes ranging from a few minutes to approximately two days at the largest dataset sizes. LLaMA experiments use 8 GPUs with data parallelism, and have maximum runtimes of under two days.

