# OpenReview forum: "Deriving Neural Scaling Laws from the Statistics of Natural Language"
_ICML.cc/2026/Conference — ICML 2026 regular_

### Official Review · Reviewer_dzfr · 2026-02-17

**Soundness:** 3
**Presentation:** 3
**Significance:** 4
**Originality:** 3
**Overall Recommendation:** 5
**Confidence:** 4

**Summary:**

This paper develops the first quantitative theory that predicts the
exponents of data-limited neural scaling laws for modern large
language models trained on natural language datasets.  The authors
identify two key statistical properties of language that determine
these exponents: the decay of pairwise token correlations with
temporal separation and the decay of next-token conditional entropy
with context length.  From these measurable statistics, they derive a
simple formula that predicts scaling exponents from first principles,
without free parameters or synthetic data assumptions.  The theory
implies that the autoregressive test loss follows a power law whose
exponent is fully determined by the language-specific entropy and
correlation decay rates.  Experiments on GPT-2– and LLaMA-style
models trained from scratch on TinyStories and WikiText show strong
agreement between the predicted and observed scaling behavior.

**Compliance With Llm Reviewing Policy:**

Affirmed.

**Key Questions For Authors:**

- Could the authors validate the theory on controlled synthetic or
surrogate datasets where γ and β can be independently manipulated,
in order to more directly test whether the predicted scaling exponent
holds beyond natural language corpora?

-Is the observed “universality” truly architecture-independent, or
does it instead reflect a shared level of representational
sufficiency?  In other words, does the scaling exponent remain
invariant across fundamentally different model classes, or only among
architectures that are powerful enough to fully exploit the data’s
statistical structure (i.e., satisfy the fast within-horizon learning
condition)?

**Limitations:**

While the paper explicitly acknowledges scale-related limitations, the
discussion remains relatively high-level and does not deeply analyze
which core theoretical assumptions might fail at industrial scale.

**Strengths And Weaknesses:**

The theoretical framework is exceptionally clean and conceptually
well-organized, deriving the data-limited scaling exponent directly
from two measurable statistical properties of natural language. The
logical progression—from entropy and correlation decay to a
prediction horizon and finally to the scaling law exponent—is both
transparent and mathematically coherent. Moreover, the predicted
exponent matches empirical measurements across datasets and
architectures with striking quantitative accuracy. This strong
agreement, achieved without free parameters, substantially strengthens
the credibility and impact of the theory.

A limitation of the study is that the theoretical predictions are
validated only at academic-scale dataset sizes, and it remains unclear
whether the proposed scaling relation continues to hold in truly
large-scale (e.g., trillion-token) training regimes.

---

> ### Author Rebuttal · Authors · 2026-03-31
>
> # Key Questions For Authors
>
> ## On validating the theory on controlled synthetic datasets
>
> In the Random Hierarchy Model (a synthetic language inspired by context-free grammar) the power-law decay of correlations and conditional entropy with context length can be explicitly computed in terms of the model parameters and varied independently. The predicted scaling exponents were found to match the empirically learning curves of deep transformers in [2] (See Fig. 3) https://arxiv.org/abs/2505.07070. **We will emphasize these references more in the revised manuscript**, as they provide a microscopic model where our assumptions can be controlled precisely.
>
> [1][Towards a theory of how the structure of language is acquired by deep neural networks](https://arxiv.org/abs/2406.00048)
>
> [2] [Scaling Laws and Representation Learning in Simple Hierarchical Languages: Transformers vs. Convolutional Architectures](https://arxiv.org/abs/2505.07070)
>
> ## On the observed universality
>
> The latter---the scaling behavior may be invariant among architectures that are powerful enough to fully exploit the data’s statistical structure. As discussed in Sec. 3, we expect simpler methods such as shallow neural networks or kernel methods to display much smaller scaling exponents, possibly suffering from the curse of dimensionality. While there are strong indications for this property from synthetic models of data (such as, again, the Random Hierarchy Model), establishing it empirically would require further work, which we leave to future projects.
>
> # Limitations
>
> We have considered TinyStories and Wikitext, two datasets that are rather homogeneous in their themes as well as their structure. Moreover, documents are typically 100-200 tokens long, limiting the benefits of exploring larger context windows. Moving towards industrial scales, we expect that the heterogeneity of the datasets, including the distribution of document sizes, could impact the behaviour of correlations and conditional entropies. **We will discuss this point further in the revised Limitations section.**

---

> > ### Author Rebuttal · Reviewer_dzfr · 2026-04-04
> >
> > I thank the reviewers for their rebuttal. I'll keep my positive score.

---

### Official Review · Reviewer_b6i4 · 2026-02-19

**Soundness:** 3
**Presentation:** 3
**Significance:** 3
**Originality:** 3
**Overall Recommendation:** 5
**Confidence:** 4

**Summary:**

In this paper, the authors derive, based on several simplifying assumptions, formulas that predict the scaling law exponents in terms of the decay of pairwise correlation of tokens and the decay of entropy with the context window length, in the regime where the model can fully exploit all existing data.
The authors also did experiments to verify their theoretical predictions.

**Compliance With Llm Reviewing Policy:**

Affirmed.

**Final Justification:**

Overall a good paper. Part of the derivation is not very rigorous, but the experiments complement the theory well, and the authors have further justified/clarified the derivation in the rebuttal.

**Key Questions For Authors:**

* Can you justify Eq. (23) (when $x \gg 1$)?
* I can understand that the noise being smaller than the largest eigenvalue is a reasonable *necessary* condition for the model to leverage the context window,
  but it seems that being smaller than the smallest non-trivial eigenvalue is a more reasonable criterion for determining if the model can fully exploit data.
  * What will happen if we use the smallest non-trivial eigenvalue (or the median of the non-trivial eigenvalues) instead of the operator norm?
  * Does it still follow a power law (like in Fig. 3)?
  * What will the predicted scaling law exponent look like in this case?

**Limitations:**

This work is mostly theoretical, and I do not see any potential negative societal impact, but the authors should make the assumptions made in the derivation more transparent.

**Strengths And Weaknesses:**

### Strengths
Overall, this is a well-written paper that contains interesting observations.
* The derived formulas are clean and depend only on statistics of the language itself.
* While I am not fully convinced by the semiformal derivation in Appendix A, the theoretical results themselves match the experimental results surprisingly well (Fig. 1 and Sec. 4).  In addition, the authors do put in a fair amount of effort to justify the assumptions that the entropy and the (operator norm of the) correlation matrix have power law decay using experiments (Sec. 4.1).
* I can see this work leading to many theoretical follow-up papers.

### Weaknesses
I do not see any major issues, but I do have several complaints about the rigorousness of the derivation and the wording.
More specifically, the authors advertise the scaling law as depending only on certain language statistics, but in the derivation, this relies on certain assumptions on the differential loss $\Delta_n$.
* First, for the $\gamma / 2\beta$ part of the exponent, it is not clear to me whether the $n$-gram loss $L_n$ will be close to the corresponding entropy when $n$ is somewhat large and $P$ is $\mathrm{poly}(n) \gg P_n^*$ (so it still passes the resolution threshold) instead of $P \to \infty$. This might be the more practically relevant regime, as the context window used in practice is definitely not only constant large, and the data is also not infinite. In this regime, probably both the model capacity and the effectiveness of the training algorithm will affect the scaling law.
* For the $\delta$ part of the exponent, I do not see why we should expect $\Delta_n$ to satisfy the specific form in Eq. (23) when $x \gg 1$, and the authors provide little justification for this formula. Eq. (23) essentially assumes that the differential loss is independent of the model capacity and training algorithm.

### Typo
* line 180: "(Ronen & Yuanzhi, 2023)". These are the first names of the authors.

---

> ### Author Rebuttal · Authors · 2026-03-31
>
> # Weaknesses
>
> ## On the $\gamma/(2\beta)$ part of the exponent
>
> We would first like to clarify that our claim is not that the $n$-gram loss $\mathcal{L}_n$ is already close to the corresponding entropy $H_n$ once $P > P_n^{\*}$. The relevant condition is instead that, beyond the resolution threshold, the excess loss $\mathcal{L}_n - H_n$ decays with $P$ faster than the error due to the finite time horizon alone $P^{-\gamma/(2\beta)}$. As Eq. 38 shows, this corresponds to the condition $\delta > \gamma/(2\beta)$, under which the overall exponent is governed by time-horizon growth rather than within-horizon learning.
>
> Whether this condition holds in practice for a given architecture is therefore an empirical question, which we test directly. In Fig. 6 (right), all fitted $\delta_n$ for $n \leq 12$ are larger than $\gamma/(2\beta)$ for the architecture studied.  As we already pointed out in the manuscript,  Eq. 23 is expected to break down in simpler methods such as kernel methods and shallow networks,  likely to suffer from the curse of dimensionality.
>
> Our work is not aimed at the $P \to \infty$ regime, but instead the practically relevant regime in which $n^{\*}(P)$ remains below the context limit $T$ while within-horizon learning is fast w.r.t. P. In fact, a larger $T$ expands the range over which this regime holds, since the condition $n^*(P)<T$ remains valid for a broader range of sample sizes $P$.
>
> ## $\delta$ part and Eq. (23)
>
> We also agree that Eq. 23 is an ansatz and should be presented as such. The precise power-law form is not necessary for the main conclusion. What is needed is only that the summed within-horizon excess decays at least as fast as the horizon-growth term $P^{-\gamma/(2\beta)}$. Eq. 23 is a sufficient assumption under which this condition reduces to $\delta > \gamma/(2\beta)$. That said, the power-law decay of $\mathcal{L}_n(P) - H_n$ measured empirically in Fig. 6 (middle/right) is consistent with this ansatz.
> Finally, we agree that the role of model capacity and training algorithm should be discussed more explicitly. Our key assumption is fast learning within a data-dependent time horizon. We do not claim that Eq. 23 holds independently of architecture or optimization. Instead, for a given sufficiently expressive model class, fast learning implicitly requires sufficient capacity and effective training. This is precisely how our experiments are conducted: For each setting, we increase model size and tune the other hyperparameters until no further improvement in test loss is observed.
>
> # Question on smallest eigenvalues
>
> As discussed in the response to reviewer LvYK, we use the top singular value as a measure of the strongest predictive signal available at lag $n$. This is because, as stated above, we want to disentangle the improvement in loss due to capturing a longer range of correlations (i.e. increasing the prediction time horizon) from the improvement due to capturing more structure on a fixed time window. Let us also remark that, since we consider a setup where P increases while V stays fixed, considering the first few singular values instead of the first would yield the same scaling. This is confirmed by the observation that the Frobenius norm of correlations displays a similar scaling to that of the top singular value.
>
> While it is reasonable to assume that the largest signals are important for learning, the very small non-trivial singular values are unlikely to play a significant role, since, by definition, there is little associated signal to exploit. For example, if $\mu$ is a very rare token, the correlation matrix $C_{\mu,\nu}$ will be small for all $\nu$. This implies that a mode localized on $\mu$ will project on very small eigenvalues. In other words, the small eigenvalue part of the spectrum will include extremely rare tokens, which can only have a limited effect on performance.

---

> > ### Author Rebuttal · Reviewer_b6i4 · 2026-04-03
> >
> > I thank the authors for the clarification and will maintain my positive score (provided that the authors will incorporate these clarifications into the revision).

---

### Official Review · Reviewer_LvYK · 2026-03-07

**Soundness:** 2
**Presentation:** 4
**Significance:** 3
**Originality:** 4
**Overall Recommendation:** 5
**Confidence:** 4

**Summary:**

The paper studies the data-limited neural scaling law and presents a theory of its origin from raw data statistics alone. They identify two dataset quantities that control the loss power law: the decay exponent of the two-point function with the token distance ($\beta$) and the decay exponent of the dataset's internal entropy as a function of the context length ($\gamma$). Notably, the theory is validated with full transformers and real natural language datasets.

**Compliance With Llm Reviewing Policy:**

Affirmed.

**Final Justification:**

The authors' rebuttal has been partially effective in addressing my concerns regarding the submission. As I indicated in my review, I believe more work could have been done to improve the robustness of the results, both on the theoretical and experimental fronts.

Overall, I do believe this is a strong submission. It has substantial strengths that significantly outweigh the weaknesses, and I would be happy to see this work presented at ICML.

**Key Questions For Authors:**

See weaknesses, especially the first point.

**Limitations:**

yes

**Strengths And Weaknesses:**

### Strengths

- The work studies the origin of scaling laws in realistic LLMs. Many previous theoretical works study the origin of scaling laws in far simpler models.
- The work includes relatively large-scale experiments that largely confirm the theory.
- The theory is simple and elegant, and explains why neural scaling laws are observed to be largely architecture independent.
- The collapse observed in figure 1 top right, and figure 4 top right is highly non-trivial.

### Weaknesses

- The paper should be precise in the derivation of the relation $n^\* (P) \sim P^{1/(2 \beta)}$, which is at the core of the contribution. The manuscript is rather loose around equation 4, and I am not sure it compares the right quantities. To the best of my understanding, equation 4 implicitly assumes the empirical covariance and the population covariance are related by additive noise, where the noise shrinks like $P^{-1/2}$ and reveals eigenvalues of the population covariance. The problem is that sampling noise in multiplicative, e.g., for the Gaussian case $\hat{C} = C^{1/2} W C^{1/2}$ for $W$ a Wishart matrix. This fact is widely recognized in the literature on scaling laws in linear models (see e.g. [1] for a very explicit discussion). These works show that eigenmodes are resolved according to a non-trivial scaling law with $P$, which translates to a non-trivial scaling law for the loss.

  While the $1/2$ scaling stated above Eq.4 is clearly justified (can be formalized by matrix Bernstein inequality), I am not sure it is the relevant one, emphasizing again that sampling noise is multiplicative. Additionally, there are other competing scales here, e.g., the fluctuation of the largest eigenvalue is expected to be Tracy-Widom rather than CLT.

  I think the manuscript would benefit greatly from a more precise discussion explaining why these are the relevant quantities to compare and where their scaling comes from.

- Some relatively straightforward tests that I expected to see in such a paper do not appear. This casts some doubt about the results, and I believe their inclusion would strengthen the submission. These include:
   - At a given $P$ and dataset, the theory suggests the model has a limited effective prediction time horizon $n^\*$. Can the tokens with $n>n^\*$ be masked without affecting the performance? Showing this is the case would be very strong evidence in favor of the suggested theory.
   - Does a large pre-trained model (e.g. large variant of Llama) give the same data intrinsic exponent $\gamma$ as the one found by the authors by training from scratch? That should be a relatively cheap test that would show the $\gamma$ found here is indeed a property of the dataset. This is a much cleaner setting that removes any confounding effects (currently, the scaling laws are predicted for the same models that are used to extract the exponents).
- There is no quantitative comparison between the predicted data-limited scaling laws and the measured ones. I would have expected to see a comparison, including confidence intervals/errors.
- On the same topic, how do the measured and predicted exponents compare with the data-limited exponents measured by canonical works in the field? Again, I expect a quantitative comparison including confidence intervals. Are the results in tension or in agreement?

If these issues can be resolved, I would be happy to recommend acceptance / strong acceptance.



[1] Atanasov A, Zavatone-Veth J A and Pehlevan C 2025 Scaling and renormalization in high-dimensional regression Online: [http://arxiv.org/abs/2405.00592](http://arxiv.org/abs/2405.00592)

---

> ### Author Rebuttal · Authors · 2026-03-31
>
> ## derivation of the relation $n^{\*}(P)\sim P^{1/(2\beta)}$
>
> We thank the reviewer for this comment. We will clarify the justification of  Eq. (4)/(26).
>
> Our setting is different from the standard Marchenko-Pastur setup considered in much of the linear-model scaling-law literature. There, one studies the spectrum of a sample covariance matrix built (in our notation) from $P$ i.i.d. observations of a single $V$-dimensional random vector, and asks how the bulk and individual eigenmodes are distorted or resolved under multiplicative sampling noise. By contrast, here we study, for each time lag $n$, the token-token covariance matrix (Eq. (25) in the manuscript)
>
> $C_{\mu,\nu}(n)=\mathbb P ( X_i=\mu, X_{i+n}=\nu ) -\mathbb P(X_i=\mu)\mathbb P(X_{i+n}=\nu),$
>
> and we use its top singular value as a measure of the strongest predictive signal available at lag $n$ (rather than attempting to characterise the full spectrum).
>
> Accordingly, the relevant statement is a detectability bound: if $\widehat C_P(n)$ is the empirical estimate of $C(n)$, then the error on each singular value is bounded by the operator norm of the estimation noise,
>
> $|\hat \sigma_k(n)-\sigma_k(n)| \le \|\widehat C_P(n)-C(n)\|_{\mathrm{op}}.$
>
> Therefore, a sufficient condition for the top singular mode at lag \(n\) to be detectable is that $\|C(n)\|_{\mathrm{op}}$ exceeds the operator-norm scale of the sampling noise. The $P^{-1/2}$ scaling entering Eq. (26) is precisely this operator-norm estimation scale (which can be justified, e.g., by standard matrix concentration bounds such as matrix Bernstein), and this is the quantity relevant for our problem of detecting the lag-$n$ signal.
>
> This also explains why Tracy–Widom fluctuations are not the relevant object here, as Tracy–Widom describes the fluctuations of the top eigenvalue of the pure-noise spectral edge around its mean. Our criterion instead concerns when the signal singular value of $C(n)$ rises above the mean operator-norm estimation error of the empirical lag-$n$ covariance. For this detectability question, the leading $P^{-1/2}$ scaling of the operator-norm noise level is the relevant scale entering $n^*(P)$, whereas Tracy–Widom governs finer edge fluctuations of the noise spectrum.
>
> **We will clarify these points in the revised manuscript.**
>
> ## Further tests, masking tokens
>
> We agree that testing the effect of finite context size is an important aspect of the theory. An explicit masking experiment would indeed be interesting; however, the main prediction of our theory is not that, at fixed parameters, a model trained with full context should make identical token-by-token predictions after masking beyond $n^{\*}(P)$. Instead, the theory predicts that in the data-limited regime, context beyond the effective horizon $n^*(P)$ should not materially affect the scaling of the loss. This prediction is already tested in the paper: the $L_n(P)$-v-$n$ curves precisely characterize how limiting context limits performance, and the collapse of these curves is the strongest support to our theory.
>
> ## Further tests, entropy exponent
>
> It is indeed important to show that the exponent \(\gamma\) reflects a property of the dataset rather than the model used to estimate it. However, using a large off-the-shelf pretrained model is not a clean measurement. The reason is that the conditional entropy $H_n$ is a property of the target data distribution. In our setup, it is estimated by training models on the dataset under study until their $n$-gram losses approach the corresponding conditional entropies. A pretrained model such as Llama has been trained on a different data mixture, and therefore, its predictions reflect that training distribution. As a result, the $n$-gram losses of such a model are not a direct estimator of the intrinsic entropy decay of the datasets considered in this paper.
>
> A cleaner test of data-intrinsicity is to keep the dataset fixed and vary the model class. This is already the logic of Fig. 2, where $\gamma$ is shown to be consistent across several transformer architectures trained from scratch on the same dataset. To strengthen this point, we also trained non-transformer-based language models on the same datasets—specifically, $n$-gram models and selective state space models—and we found the same entropy-decay exponent $\gamma$. (see Entropy_TinyStories_ figure of [THIS LINK](https://anonymous.4open.science/r/Deriving-Neural-Scaling-Figures-8ECF/)). This provides substantially stronger evidence that $\gamma$ is a property of the dataset rather than of a specific architecture or training recipe.  **We will add and discuss these curves** in the rightmost panels of Figs. 2 and 5 of the revised manuscript
>
> ## Quantitative comparison and confidence intervals
>
> **3. Quantitative comparison between predicted and measured exponents.**
> See response to referee QNVf.
>
> ## Comparison with the literature
>
> Please see the response to Key Question number 3 of reviewer QNVf.

---

> > ### Author Rebuttal · Reviewer_LvYK · 2026-03-31
> >
> > Dear authors,
> >
> > While my questions have been mostly substantive-scientific, your response has largely been presentation-clarification-like, which feels a bit like we might be talking past each other.
> >
> > **W1**. I'm simply calling out that the manuscript and your rebuttal seem to assume additive noise implicitly (you call $\widehat C_P(n)-C(n)$ the sampling noise). I gave you an example where this noise is multiplicative; your response has largely been "this is a different setting" instead of explaining why you believe this different setting gives rise to a noise of a different character. If you could, I ask that your explanation be on a quantitative, mathematical basis rather than conceptually-different-setting basis. I don't believe there is a need to reiterate what already appears in the manuscript.
> >
> > Put it a different way: You describe eigenvalues that emerge out of a noise floor: what is the source of this noise? In your setting, you have no explicit noise added to the samples.
> >
> > **W3.** A quantitative comparison is needed here. The plot is not quantitative, but it could be: can you quantify the deviations?
> >
> > I think you currently somewhat underplay the importance of this request. The hypothesis presented in this paper could, potentially, already be ruled out statistically based on the data you have. Fig.1, bottom, does not show perfect agreement; judging whether this agreement should still be taken as good or not crucially requires confidence intervals.
> >
> > - This should be straightforward to report and is important for assessing the submission: **Please give the confidence bounds of your fits, as given by scipy or whatever other program you use.**
> >
> > I strongly urge the authors to include these numbers in their response / in an anonymous link.
> >
> > **W4.** I couldn't find your response.
> >
> > **Regarding measuring intrinsic entropy using a pretrained LLM.**
> > I think you are making a claim here that is non-obvious from your submission. I understood the intrinsic entropy to be minimal and irreducible; if a large pretrain LLM can do better, it is neither. Noting the obvious: the predictions of Llama are context-dependent, and different corpora have different entropy/perplexity.
> > - How should one think of your intrinsic entropy of a corpus compared to measuring the perplexity of it with a well-trained model?
> > - If you think of these as very different concepts, I think you should clarify this in the text. Are they at all related? I maintain that comparing them is still valuable, but please prioritize responding to my first two points.

---

> > > ### Author Response · Authors · 2026-04-03
> > >
> > > **W1**: Let us summarise our reasoning behind the derivation of $n^*(P)$ and answer the specific questions of the referee. We will clarify these points in the revised manuscript.
> > >
> > > We define $n^\*(P)$ as the context scale below which the token-token correlation matrix $C(n)$ can be reconstructed from a finite sample of size $P$. We look at such correlations because, in synthetic hierarchical languages, they were shown to contain the signal that allows a learner to reconstruct the latent hierarchical structure of the data (ref. (Cagnetta \& Wyart, 2024) of our paper). The derivation of $n^*(P)$ is essentially a signal-to-noise argument: a learner with $P$ training data has access only to the empirical estimate $\widehat{C}_P(n)$. **Therefore, the noise is not explicitly added to the samples, but is rather a sampling noise $\widehat{C}_P(n)-C(n)$, additive by definition.** This noise controls, by Weyl's theorem, the error in estimating the singular values of $C(n)$, as well as the fluctuations of the individual elements of the matrix (as shown below).
> > >
> > > Since $\widehat{C}_P(n)$ contains frequencies of co-occurrence, the nature of the noise is readily obtained with the central limit theorem. Given the noise, the scaling of $n^*$ with $P$ can be formalized with either the Frobenius norm or the operator norm, which empirically display the same scaling with $n$. This scaling was also tested empirically in Cagnetta \& Wyart (2024), Fig.~4, top right.
> > >
> > > Let
> > > \\[
> > > C(\mu,\nu)=P(\mu,\nu)-P(\mu)P(\nu)
> > > \\]
> > > be the true $V\times V$ covariance matrix for some context $n$. For large $n$, the covariance is very small and $P(\mu,\nu)\approx P(\mu)P(\nu)$. Consider the empirical estimator
> > > \\[
> > > C^P(\mu,\nu)=\hat P(\mu,\nu)-\hat P(\mu)\hat P(\nu).
> > > \\]
> > > Its fluctuations correspond to those of the count of pairs $(\mu,\nu)$, of order $P(\mu)P(\nu)P$. In the limit of large $P$,
> > > \\[
> > > \delta C(\mu,\nu):=C^P(\mu,\nu)-C(\mu,\nu)\sim O_p \left(\frac{\sqrt{P(\mu)P(\nu)}}{\sqrt P}\right).
> > > \\]
> > > The typical noise magnitude in Frobenius norm is
> > > \\[
> > > \\| C^P-C \\|_F \sim \sqrt{\sum\_{\mu,\nu} \mathbb{E}\[\delta C(\mu,\nu)^2\]} \sim \frac{1}{\sqrt P}\sqrt{1+2 \sum\_\mu P(\mu)^2}\sim \frac{1}{\sqrt P}.
> > > \\]
> > > The signal Frobenius norm is
> > > \\[
> > > \\| C \\|_F=\sqrt{\sum\_{\mu,\nu}C(\mu,\nu)^2}.
> > > \\]
> > > The signal emerges from noise when
> > > \\[
> > > \\| C \\|_F\gtrsim \\|C^P-C\\|_F
> > > \quad\Rightarrow\quad
> > > P\gtrsim \frac{1}{\\|C\\|_F^2},
> > > \\]
> > > as claimed. Similar arguments can be made for the operator norm.
> > >
> > > **W3**: We did not report error estimates from the fit because the dominant source of uncertainty is not statistical noise, but systematic finite-size effects, as in critical phenomena. The same occurs here, and we use the corresponding toolbox to estimate errors: (i) when functional collapse breaks down, (ii) how the fitting range affects exponents, estimated by bootstrap methods, and (iii) how finite $T$ affects exponents. We are happy to provide a table including fit-based error estimates and confidence intervals. All are consistent with our theory, although confidence intervals are not in our view the most appropriate measure of the error here.
> > >
> > > | Exponent | Collapse-based estimate | Bootstrap estimate ($\pm$ SE) | Theory prediction $\hat{\alpha}_D=\gamma/(2\beta)$ | 95\% confidence interval from fit to $\mathcal{L}(P)$ |
> > > |---|---|---|---|---|
> > > | $\beta$ | $0.88 \pm 0.09$ | $0.89 \pm 0.06$ |  |  |
> > > | $\gamma$ | $0.33 \pm 0.03$ | $0.325 \pm 0.003$ |  |  |
> > > | $\alpha$ |  | $T=128: 0.16 \pm 0.01$$T=256: 0.18 \pm 0.01$$T=512: 0.19 \pm 0.01$ | From collapse: $\hat{\alpha}_D=0.19 \pm 0.03$. From bootstrap: $0.18 \pm 0.01$ | $T=128: (0.151, 0.174)$$T=256: (0.158, 0.205)$$T=512: (0.161, 0.203)$ |
> > >
> > > As the table demonstrates, the theoretical prediction is compatible with the reported confidence intervals.
> > >
> > > **W4**: Apologies, the response was already given to the first reviewer and not reported here: "We agree that this would be an interesting comparison. Unfortunately, we do not have access to the complete training dataset used by Kaplan et al. (required to measure ...) and to their best-performing models (required to measure ...).''
> > >
> > > **Conditional entropy:** This is a key point and will be clarified in the revision. Given a dataset $D$, the minimum of the cross entropy $L_n$ is the conditional entropy $H_n$. If the model has enough capacity, for a finite context $n$, $L_n$ must converge to $H_n$ as $P$ goes to infinity. We test empirically that this convergence occurs for limited $n$ when $P$ is large enough, for transformer-based and state-space language models, and even for N-gram estimators when $n$ is small enough.
> > >
> > > If a model is trained on another dataset, its cross-entropy on $D$ will be sub-optimal, so $L_n>H_n$. In practice, testing this directly is not possible because tokenisation is different. Ultimately, the convergence of our estimate of $H_n$ across very different methods—transformers of different families, state-space models, and N-grams—strongly supports our claim.

---

### Official Review · Reviewer_QNVf · 2026-03-09

**Soundness:** 3
**Presentation:** 2
**Significance:** 4
**Originality:** 4
**Overall Recommendation:** 5
**Confidence:** 4

**Summary:**

This work establishes a precise relationship between the scaling laws of the loss with respect to dataset size as a function of measurable properties of the data. More precisely, using only the decay exponent of conditional entropy of the $n+1$-th token with respect to the $n$ previous ones, as well as token-token correlation, the authors can predict the scaling exponent of loss with respect to dataset size. Additionally, they provide empirical measurements illustrating that their theoretical predictions match practical exponents.

**Compliance With Llm Reviewing Policy:**

Affirmed.

**Final Justification:**

The main strengths are originality and significance as this work takes a novel direction in explaining quantitatively the loss scaling exponents observed in practiced through measurable statistics from the data.

The main weakness in my opinion is the relative sparsity of discussion and details aroud precision of the exponents measured. In practice, even a tiny variation of a few percents in the scaling exponent can change substantially the resulting performance of a model, especially at scale, and quantitative predictions beyond that precision would be hard to use in practice.

However, the authors discussed these in their rebuttal as well as answer to other reviewers. I also don't think this is a major problem since this work is one of the first exploring that direction, and has substantial contribution.

**Key Questions For Authors:**

1) I was wondering if the authors had a chance to compute confidence intervals for the reported measured exponents. I think it is an important missing point of this analysis, which aims at confronting theoretical predictions with empirical measurements. Additionally, as a follow-up to the weakness mentioned above, I would be glad to engage in a discussion on the reported $\beta$ measurement of Figure 3, right, and on why the authors chose to measure the exponent in the left region.

2) Models are typically trained with a fixed context window, which can actually be increased during training. How do you think a fixed context window could affect your results, and in the case of an increasing context window, do you think it could lead to a change of scaling exponent (since the main driver of the exponent from your study seems to be the largest length that the model can take into account, $n^*(P)$).

3) It could be interesting to compare the reported exponents of loss vs tokens of this work with other previously reported of larger scale models, for example, from [1].

[1] Kaplan, Jared, et al. "Scaling laws for neural language models." arXiv preprint arXiv:2001.08361 (2020).

**Limitations:**

The authors adequately address some limitations of their study. I highlight below one more limitation which could be worth discussing.

This work derives scaling laws of the loss with respect to dataset size, but doesn't account for model size (finite expressivity of the model). In particular, compute optimal studies such as in [1] would not be captured in this framework, which implicitly assumes infinite model capacity and a regime only limited by data complexity.

[1] Hoffmann, Jordan, et al. "Training compute-optimal large language models." arXiv preprint arXiv:2203.15556 10 (2022).

**Strengths And Weaknesses:**

**Strengths**

*Originality* This work provides an original and novel theoretical explanation and predictions about the scaling behaviour of neural networks on real data. Of particular importance, and as opposed to most theoretical works of the field, these predictions can be compared to experiments in a precise way by measuring parameters playing a role in the theory, directly on real data.

*Significance* This work introduces a novel view and understanding of the scaling laws of large language models based on the data structure and measurable parameters. This paves the way to important research directions about better understanding the properties of real data and its relations to scaling exponents. In particular, this work contributes to bridging a non-trivial gap between theoretical predictions of scaling laws and empirical confrontation with the experiments.

**Weaknesses**

*Soundness* An important weakness of this work is the absence of confidence intervals for the measurements. The authors argue that a main contribution of this work is to provide theoretical predictions that can be compared to experimental results. However, this would require confidence interval measurements and computations. For example, it is not clear what the precision of the exponents reported in equations 10 to 14 is, which are all given with two digits. I presume the actual precision could be lower than this. A main point which attracted my attention is how the authors measured the exponent $\beta$ in Figure 3, right. This plot exhibits (as the authors notice) a very nonlinear behaviour and the authors chose to report "the exponent of the first, shortlag, stage, which is the regime relevant for our prediction". This is not justified, and goes against my understanding that under the relation $n\approx P^{1/2\beta}$ with $\beta \approx 0.9$ and $P\approx 10^7$ in Figure 4, the relevant regime would be with $n \approx 10^4$, which is on the right of the figure (and actually outside the scope of measurements of this figure). However, using the exponent on the right of this figure would change substantially the measured exponent and its closeness to experimental results.

Additionally, the analysis uses strong assumptions regarding, for example, the fact that the $E_n$ are negligible in equation 5. Note that although strong, these are overall well-discussed in Appendix A as well as Section 5.

*Presentation* The results are overall well-explained and easy to follow. However, some prior works regarding the measurements of decay of the entropy of the next token given previous tokens weren't properly cited, see eg [1]. It would be interesting to compare the reported value to the measurements the authors did using trained language models.

[1] Takahira, Ryosuke, Kumiko Tanaka-Ishii, and Łukasz Dębowski. "Entropy rate estimates for natural language—a new extrapolation of compressed large-scale corpora." Entropy 18.10 (2016): 364.

---

> ### Author Rebuttal · Authors · 2026-03-31
>
> # Weaknesses
>
> ## Absence of confidence intervals
>
> We agree that uncertainty quantification is important, especially since the paper makes quantitative comparisons. Indeterminacy here is not statistical (our test losses are averaged for four different seeds with negligible variance) but is systematic and depends mostly on the fitting range considered.  Meta-parameters, in particular the horizon $T$, also affect results, as we report.
>
> Since our theory predicts a collapse of the curves $L_n(P)$ when the axes are rescaled using the exponents $\beta$ and $\gamma$, we can estimate uncertainties following a standard recipe from the theory of critical phenomena: consider the range of exponent values where the collapse is satisfying. As documented [HERE](https://anonymous.4open.science/r/Deriving-Neural-Scaling-Figures-8ECF/) (LossCollapse figures), we find the quality of collapse to deteriorate when exponents change by about 10% for Tinystories and 15% for WikiText.
>
> Plotting these ranges of exponents $\beta$ and $\gamma$ on the curves $C(n)$ and $H(n)$ used for their measurement (L_n_ and C_n_ figures), shows that similar uncertainties would be obtained by changing the fitting range considered (as long as $n$ is not too larg, see discussion below).
>
> Finally, systematic errors on the predicted training curve exponent $\alpha$ can be estimated by considering how $\alpha$ depends on the choice of $T$, and on the fitting range considered. Here we find that uncertainties are of order 10% in relative terms.
>
> **We will add a discussion on uncertainties in exponents in the main text of the revised manuscript, and display all the figures shown in a new Appendix.**
>
> ## Exponent $\beta$ in Figure 3, right
>
> As discussed in the manuscript, our rationale for fitting the short-lag regime is that $n^*(P)$ falls into that range for the scale of $P$ we consider. This is visible in Fig. 4, top right. According to our theory, $L_n(P)$’s decay must slow down when $P$ becomes larger than $P^{\*}(n)$. Curves with $n=2^7=128$ (yellow points) do not bend significantly, thus the maximal range of $n$ where our theory is being tested is $n=32$ or $64$, where curves do bend. An equivalent way to see this is through the prefactor $A$ entering $P = A n^{2\beta}$, which the referee assumed to be $~1$. From the paragraph above, we estimate $A>10^4$ instead.  We further note that even if fitting $\beta$ using data up to $n=128$, rather than $n=32$, the resulting exponent still lies within the uncertainty range inferred from the collapse analysis (see link above). **In the revision,** we will further stress that the short-lag fit is empirically motivated: the long-lag regime may be relevant asymptotically, but it is not the regime probed by the current and dataset sizes.
>
> ## Strong assumption about the $E_n$’s
>
> The reviewer notes that “these are overall well-discussed in Appendix A and section 5”. Furthermore, we remark that *a)* the agreement between predicted exponent and scaling curves and the “highly non-trivial” (in the words of reviewer LvYK) collapse validates this assumption empirically; *b)* this assumption can be justified theoretically when studying analytically tractable models of data, such as the Random Hierarchy Model~[1,2] (see also reply to Reviewer dzfr).
>
> [1] https://arxiv.org/abs/2505.07067.
> [2] https://arxiv.org/abs/2505.07070.
>
> ## Presentation
>
> This is a very interesting comparison that **we will expand upon in the revised manuscript**. After correcting for the difference between conditional entropy and the standard sequence entropy, our assumption in Equation (6) corresponds to the formula $f_1(n)$ of the cited reference, with $\gamma=1-\beta$. Despite the reference considering a character-level tokenisation, they report $\beta \sim 0.8$, corresponding to $\gamma~0.2$, not too far from the values we report. Note that, as discussed in reply to reviewer LvYK, we performed additional results on N-gram models and State space models that confirm our measure of $\gamma$.
>
> # Key Questions
>
> 1. See **Exponent $\beta$ in Figure 3, right** section.
> 2. Our results apply to a regime of a large maximal context window $T$. As $n^{\*}(P)$ approaches $T$, the exponent might change, as the performance will be limited by the finite context window $T$ instead of $n^{\*}(P)$. This is visible in Figure 1, bottom, where the loss curves start deviating from our prediction at large $P$, with those having smaller $T$ deviating earlier. **In the revision,** we will stress this point further by expanding the discussion around Eq. (5).
> 3. We agree that this would be an interesting comparison. Unfortunately, we do not have access to the complete training dataset used by Kaplan et al. (required to measure $\beta$) and to their best-performing models (required to measure $\gamma$).
>
> # Limitations
>
> We will comment further on the possibility of studying the effect of model size in the Conclusions and Limitations section of the **revised manuscript**.

---

> > ### Author Rebuttal · Reviewer_QNVf · 2026-04-04
> >
> > I thank the authors for their detailed answers to my questions. My main concerns were resolved and I will increase my score to recommend acceptance mostly because of this work's originality and that it could spark very interesting new research directions around quantitatively understanding loss exponents from summarized data measurement. The following concerns have been resolved:
> >
> > **Exponent $\beta$ in Figure 3, right**
> > I thank the authors for their explanation and details about which regime is of interest for the exponent measurement.
> >
> > **Confidence intervals and comparison with other exponents from the litterature**
> >
> > The authors answered that "Indeterminacy here is not statistical" but systematic which I would agree with. However, this is the reason why it would have been interesting to see a more detailed comparison with loss scaling exponents from the litterature. I understand the authors reply that the datasets are not necessarily available which makes the measure of the model's parameters hard, but it would be interesting to have an idea of the range they evolve in. For example I appreciate that the authors discussed the related work "$\beta \sim 0.8$, corresponding to $\gamma~0.2$, not too far from the values we report"  but it still highlights non negligible variations in the exponents measured.

---

### Decision · Program_Chairs · 2026-04-30

**Decision:**

Accept (regular)

**Comment:**

This work is a breath of fresh air in the "neural scaling laws" space. The authors derive Kaplan-type neural scaling laws directly from statistics of the input data. They show that the scaling law exponents are completely characterized by (a) decay rate of pairwise token correlations with time separation between token pairs,and (b) decay rate of the next-token conditional entropy with the length of the  conditioning context. Extensive experiments on small language models (GPT 2, etc) and realistic datasets (wikitext, tinystories, etc.)

There is clear concensus among the reviewers regarding the excellent quality of the work. I recommend acceptance.

Important missing references:

- Cabannes et al. (ICLR 2024) "Scaling laws for associative memories"
- Łukasz Dębowski (2025) "From Zipf's Law to Neural Scaling through Heaps' Law and Hilberg's Hypothesis"